# Mechanism of millisecond Lys48-linked poly-ubiquitin chain formation by cullin-RING ligases

Joanna Liwocha [1,5], Jerry Li [2,5], Nicholas Purser[2], Chutima Rattanasopa[2], Samuel Maiwald[1], David T. Krist[1], Daniel C. Scott[3], Barbara Steigenberger[4], J. Rajan Prabu [1], Brenda A. Schulman [1,6] ✉ & Gary Kleiger [1,2,6] ✉

E3 ubiquitin ligases, in collaboration with E2 ubiquitin-conjugating enzymes, modify proteins with poly-ubiquitin chains. Cullin-RING ligase (CRL) E3s use Cdc34/UBE2R-family E2s to build Lys48-linked poly-ubiquitin chains to control an enormous swath of eukaryotic biology. Yet the molecular mechanisms underlying this exceptional linkage specificity and millisecond kinetics of poly-ubiquitylation remain unclear. Here we obtain cryogenic-electron microscopy (cryo-EM) structures that provide pertinent insight into how such poly-ubiquitin chains are forged. The CRL RING domain not only activates the E2-bound ubiquitin but also shapes the conformation of a distinctive UBE2R2 loop, positioning both the ubiquitin to be transferred and the substrate-linked acceptor ubiquitin within the active site. The structures also reveal how the ubiquitin-like protein NEDD8 uniquely activates CRLs during chain formation. NEDD8 releases the RING domain from the CRL, but unlike previous CRL–E2 structures, does not contact UBE2R2. These findings suggest how poly-ubiquitylation may be accomplished by many E2s and E3s.

The enzymatic assembly of poly-ubiquitin chains onto protein substrates is a defining feature of eukaryotic cell biology. Ubiquitin chain formation determines the fates of substrates; for instance, by affecting the cellular localization of the modified protein, or in many cases, eliciting degradation by the 26S proteasome[1]. Poly-ubiquitin chains are forged during the covalent joining of a ubiquitin bound to a protein substrate with an enzyme-activated ubiquitin. Ubiquitin is a highly conserved protein containing seven lysine residues that all may serve as linkage points for poly-ubiquitin chains[2]. Nonetheless, Lys48 appears to be of particular importance, as it is the only ubiquitin lysine residue required for viability in yeast[3] and has consistently been identified as the most abundant poly-ubiquitin chain linkage type in cells derived from a variety of organisms[4], including humans[5]. As such, it is of great importance to elucidate the enzymatic mechanisms of Lys48-specific poly-ubiquitin chain formation.

Cdc34/UBE2R-family E2s are highly specialized in forging ubiquitin chains with Lys48-linkages that elicit 26S proteasome-dependent protein degradation[2,6]. Yeast Cdc34 was amongst the first components of the ubiquitin system identified[7], and its essentiality for the cell division cycle helped guide the discovery of its partner E3s as CRLs[8–10]. CRL and Cdc34/UBE2R-catalyzed poly-ubiquitylation control gene expression, metabolism, signaling, protein trafficking, targeted protein degradation and more[11,12]. Therefore, it is important to understand how CRLs, together with UBE2R-type E2s, poly-ubiquitylate substrates.

CRLs are a large family of modular multi-subunit complexes, with approximately 300 members in humans. CRLs recruit substrate

[1]Department of Molecular Machines and Signaling, Max Planck Institute of Biochemistry, Martinsried, Germany. [2]Department of Chemistry and Biochemistry, University of Nevada, Las Vegas, Las Vegas, NV, USA. [3]Department of Structural Biology, St. Jude Children's Research Hospital, Memphis, TN, USA. [4]Mass Spectrometry Core Facility, Max Planck Institute of Biochemistry, Martinsried, Germany. [5]These authors contributed equally: Joanna Liwocha, Jerry Li. [6]These authors jointly supervised this work: Brenda A Schulman, Gary Kleiger. ✉e-mail: schulman@biochem.mpg.de; gary.kleiger@unlv.edu

'degron' motifs that bind to a specific substrate receptor module. Numerous distinct substrate receptors bind interchangeably to core cullin-RING modules. For example, F-box proteins are substrate receptors that function with the cullin-RING module CUL1–RBX1. Foundational F-box proteins include FBXW7 and βTRCP, which control the degradation of numerous oncoproteins, such as cyclin E and c-Myc, or β-catenin and IκBα, respectively[13]. Meanwhile, BC-box proteins are substrate receptors that function with CUL2–RBX1 (ref. [14]). Well-characterized BC-box proteins include VHL and FEM1C, which regulate hypoxia-induced degradation of HIF1α[15] or recognize protein carboxyl termini as degrons in the nucleotide exchange factor SIL1 (refs. [16–18]), respectively.

To poly-ubiquitylate their substrates, CRLs must partner with ubiquitin-carrying enzymes, which typically specialize in either 'priming'—that is, directly modifying substrate—or 'extending' ubiquitin chains. Notably, Cdc34/UBE2R-family E2s are specialized in the latter category and add ubiquitin molecules to primed CRL substrates at a millisecond time scale[19]. This ultra-rapid formation of Lys48-linked chains presumably drives the timely degradation of CRL substrates. These ubiquitylation events are activated by NEDD8 modification of the cullin subunit[20]. Although structures with various substrates and distinct ubiquitin-carrying enzymes have defined how NEDD8-modified, CUL1-based CRLs catalyze priming[21,22], the mechanisms of poly-ubiquitylation remain elusive.

Prior studies have suggested that CRL-catalyzed ubiquitin chain formation with UBE2R-family E2s differs from the well-studied priming reactions; for example, as mediated by E2s in the UBE2D family. For instance, CRLs are the only genetically validated E3 partners of Cdc34 and UBE2R-family E2s[23,24]. By contrast, UBE2D-family E2s are exceptionally promiscuous and function with a large fraction of all E3s characterized to date[25]. A second difference concerns the hallmark feature of most ubiquitin ligases, the RING domain[26,27], which serves as the catalytic entity of CRLs. RING domains typically function by allosterically activating ubiquitylation by facilitating close contact between E2 and its covalently bound ubiquitin (hereafter E2-ubiquitin, where the '-' represents or mimics an activated state, or donor ubiquitin). Unlike UBE2D-family E2s, Cdc34/UBE2R-family E2s activate donor ubiquitin in the so-called 'closed' conformation even in the absence of E3 (refs. [28,29]). Cdc34/UBE2R can thus forge unanchored ubiquitin chains in an E3-independent manner, albeit relatively slowly[30]. Ubiquitin chain formation by Cdc34/UBE2R-family E2s is accelerated by one to two orders of magnitude in the presence of a CRL[31]. This demonstrates the crucial catalytic role of the E3 but raises the question of how this is achieved.

Amino acid sequences of the Cdc34/UBE2R-family are unique amongst E2s, with a 16-residue insertion in the catalytic UBC domain and an acidic C-terminal tail[32–39]. The insertion is essential for yeast viability[36,37] and achieving millisecond rates of poly-ubiquitylation[33], but its function is not explained by any prior structure. The acidic tail dynamically binds to a basic canyon on the cullin[40,41] and helps to form the closed conformation with donor ubiquitin even in the absence of E3 (ref. [29]).

Another feature that remains the subject of debate is the function of the ubiquitin-like protein NEDD8, the primary activator of CRLs in cells. NEDD8 activates CRL substrate priming by directly binding a UBE2D-ubiquitin[22,42]. NEDD8 also activates ubiquitin chain extension and reduces the Michaelis–Menten constant ($K_m$) of UBE2R-family E2s[43]. However, deletion of the domain containing the neddylation site on CUL1 stimulated the activity of a UBE2R-family E2 though in a crude system[44], implying differences in NEDD8 function during priming and chain extension.

To date, no human RING-based E3 had been visualized during poly-ubiquitylation of an E3-bound substrate to promote Lys48-specific ubiquitin chains or any other chain type. Given the biological importance of Cdc34/UBE2R E2s and CRLs and owing to their unique and yet perplexing catalytic elements, we determined the structural basis for their millisecond production of Lys48-linked ubiquitin chains.

## Results

### Cryo-EM structure showing CRL substrate poly-ubiquitylation

We sought to determine a cryo-EM structure of a neddylated CRL[20] with UBE2R2-ubiquitin poised to modify a ubiquitin-primed substrate. Given that structure determination is an empirical endeavor, several distinct CRL complexes were tested. The highest resolution maps were obtained with CRL2[FEM1C] (containing neddylated CUL2–RBX1 and substrate receptor Elongin B/C–FEM1C[16–18]; Fig. 1a). As the transition state for poly-ubiquitylation is fleeting, a ligation mimic was used to join acceptor ubiquitin fused to a C-terminal degron peptide substrate (Sil1) with donor ubiquitin and eventual cross-linking to the UBE2R2 active site (Fig. 1b and Extended Data Fig. 1a). The structure resolved to 3.8 Å resolution (Fig. 1c–e, Extended Data Figs. 1 and 2, Table 1 and Supplementary Video 1) and enabled rationalization of poly-ubiquitylation on the millisecond time scale.

The individual components observed in previous structures matched expectations. For example, (1) the amino-terminal side of CUL2 binds the substrate receptor complex, while its C-terminal side interacts with the RBX1 subunit harboring the RING domain that recruits E2s[45] (Fig. 1c); (2) the UBE2R2 catalytic UBC domain and C-terminal extension both participate in forming the closed conformation for the UBE2R2-ubiquitin conjugate[28,29,46–49]; and (3) the acceptor ubiquitin and the UBE2R2 UBC domain interact (Fig. 1e). Interestingly, NEDD8 is not visible in the cryo-EM map.

Despite the seeming similarity to prior structures, there are two striking differences that explain how CRLs activate extension of Lys48-linked ubiquitin chains. First, the conformation of the UBE2R2-ubiquitin intermediate shows significant rearrangement compared to the prior crystal structure that lacked an E3 (ref. [29]) and contained an inhibitor that prevents the discharge of ubiquitin from a UBE2R-family E2 active site[50,51] (Extended Data Fig. 3a). However, the UBE2R2-ubiquitin structure here was highly similar to an E2-ubiquitin bound to RING adopting the closed conformation[47] (Extended Data Fig. 3b). Morphing between the E3-free and bound UBE2R2-ubiquitin conformations allows for visualizing CRL-dependent alignment into the activated conformation (Supplementary Video 2). These results are consistent with a model in which the inhibitor may block poly-ubiquitylation by securing donor ubiquitin against the E2 UBC domain in a closed yet inactive conformation. Second, these differences occur concomitantly with the organization of the distinctive Cdc34/UBE2R-family E2 insertion (His98–Arg113), forming a loop that is the heart of the complex. Multiple loop residues hover near the UBE2R2 active site, including His98 (which has been shown to be critical for UBE2R2 biochemical activity and for yeast viability[32]). The structure confirms the importance of His98, with its imidazole ring residing directly across from Cys93, the key active site residue that becomes thioesterified to donor ubiquitin (Extended Data Fig. 1b). Contrary to expectations, the E2 insertion, commonly referred to as the acidic loop, did not contact basic patches on neighboring subunits. Rather, the loop residues unify the E2, E3 and the ubiquitins to be adjoined into a cohesive functional unit (Fig. 1e, inset). We thus rename this insertion sequence 'synergy loop'.

### Molecular synergy promotes millisecond poly-ubiquitylation

Multiple interactions between the UBE2R2 synergy loop and RBX1 facilitate shaping the loop's conformation, which in turn forms interfaces with both the donor and acceptor ubiquitins (Fig. 2a). Conserved residues of the synergy loop radiate outward to interact with the other components involved in poly-ubiquitylation (Extended Data Fig. 3c,d and Supplementary Video 3). Additionally, at least three residues internally organize the loop conformation (Asp102, Glu108 and Arg113; Fig. 2b). To ascertain how these interactions may affect UBE2R2 activity, various mutant UBE2R2 proteins were assayed for the neddylated CRL-dependent formation of unanchored poly-ubiquitin chains[33,38] (Extended Data Fig. 3e). In brief, UBE2R2 is first thioesterified to radio-labeled donor ubiquitin (the N-terminal tag that promotes

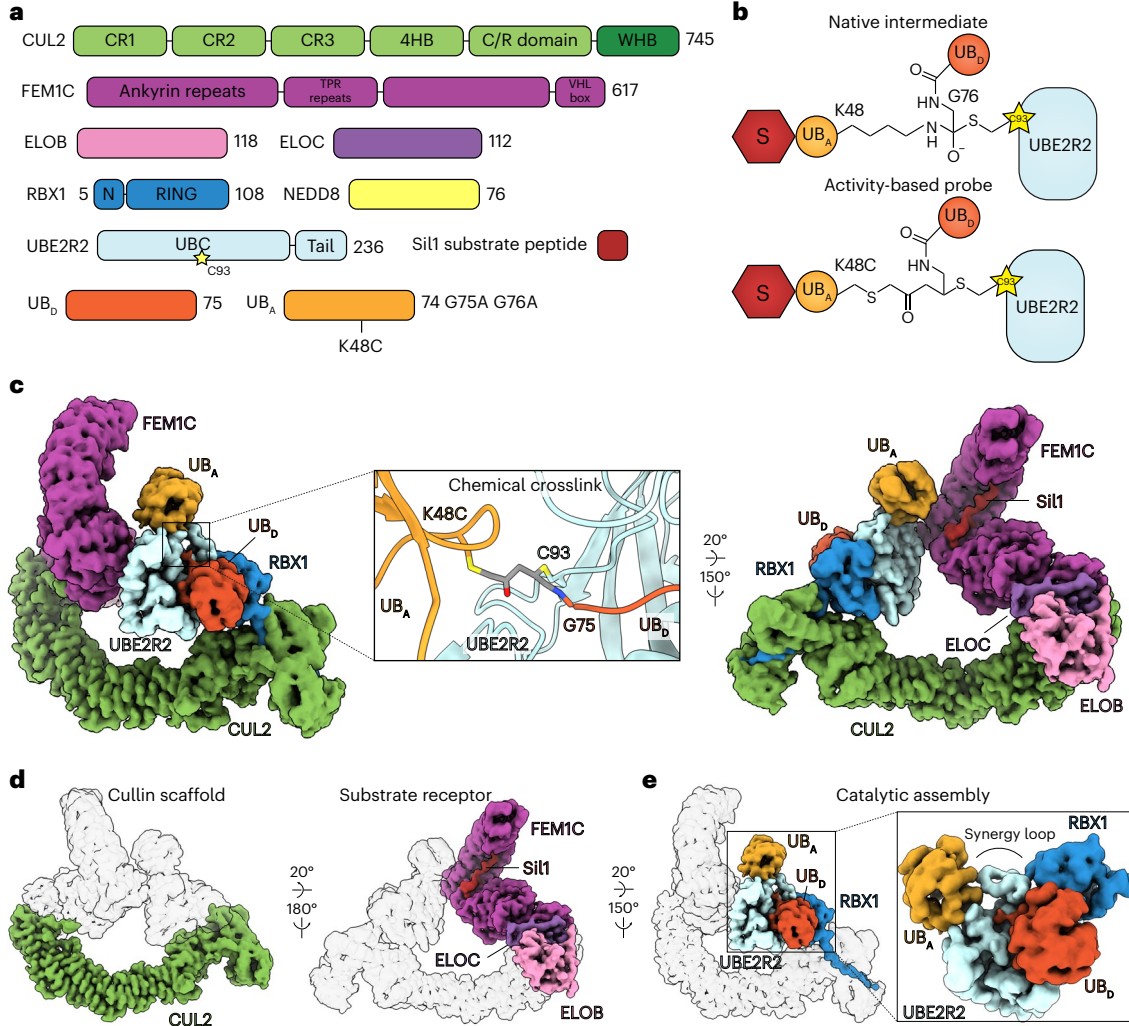

**Fig. 1 | High-resolution cryo-EM structure of a CRL promoting poly-ubiquitylation. a**, Schematic representation for all protein subunits used to form the chain formation complex for cryo-EM. **b**, Diagram comparing the native transition-state geometry for UBE2R2-dependent chaining with a stabilized ubiquitin-chain forming architecture. The ligation mimic consists of a peptide substrate (S; red hexagon) fused to a K48C acceptor ubiquitin (UB$_A$; light orange circle) crosslinked to donor ubiquitin (UB$_D$; dark orange circle), followed by reaction with the UBE2R2 active site cysteine. **c**, Various orientations of the DeepEMhancer composite cryo-EM map representing the structure. Electron density for each subunit has been colored according to the schematic in **a**. **d**, Cryo-EM maps highlighting the cullin scaffold (left) and the substrate receptor complex (right). **e**, Cryo-EM map showing the catalytic core containing UBE2R2, UB$_D$, UB$_A$ and RBX1. ELOB, Elongin B; ELOC, Elongin C.

labeling through phosphorylation was shown to not affect UBE2R2 activity; Extended Data Fig. 3f), followed by the addition of unlabeled acceptor ubiquitin and CRL. Indeed, the mutation of synergy loop residues located at each interface led to the impairment of poly-ubiquitin chain formation (Extended Data Fig. 3g).

The UBE2R2 synergy loop helps integrate the donor and acceptor ubiquitins into the catalytic conformation. The Ser106 and Glu112 in UBE2R2 point toward the C-terminal tail in the donor ubiquitin (Fig. 2c). Across from the donor, Asp103 in UBE2R2 points toward His68 in the acceptor ubiquitin (Fig. 2d), which presumably helps place Lys48 into the UBE2R2 active site. Mutation of both interfaces, as well as the loop organizing residues, resulted in higher $K_m$ values of unanchored acceptor ubiquitin for the UBE2R2-mediated poly-ubiquitylation complex (Fig. 2f, Extended Data Fig. 4a,b and Table 2).

UBE2R2's synergy loop mediates activation by the E3. A unique E2–E3 interface is formed between the synergy loop and the RBX1 RING domain, stabilized by Glu108 in UBE2R2 and Arg91 in RBX1 (Fig. 2e). Accordingly, neddylated CRL-dependent UBE2R2 activation was impaired by an R91E mutation (Extended Data Fig. 4c,d). This also weakened the apparent affinity of unanchored acceptor ubiquitin for UBE2R2 in the presence of neddylated CUL2–RBX1 (Fig. 2f, Extended Data Fig. 4b and Table 2).

UBE2R2 shares extensive sequence and functional similarity with its paralogous human E2 UBE2R1 (Extended Data Fig. 3c). To test for conservation of the interfaces observed in the structure, UBE2R1 synergy loop mutants were assayed with neddylated CRL2$^{FEM1C}$, while wild-type (WT) UBE2R1 was assayed with previously described mutations in the RING domain or donor ubiquitin (for the latter, controls were performed to ensure comparable UBE2R1 and UBE2R2 loading for WT and mutant ubiquitins; Extended Data Fig. 4e). Like UBE2R2, these mutations all resulted in significant increases in the $K_m$ values of unanchored acceptor ubiquitin for UBE2R1-catalyzed ubiquitin chain formation (Extended Data Fig. 4f and Table 2).

### Intricate placement of Lys48 into the UBE2R2 active site
The UBE2R2 catalytic UBC domain also recruits the acceptor ubiquitin through residues located on both the N-terminal portion of α-helix3 and its preceding loop in a unique manner (Fig. 3a). This differs from a UBE2R2 acceptor ubiquitin binding surface recently proposed[52]. The placement of the acceptor ubiquitin relative to the UBE2R2 UBC

**Table 1 | Cryo-EM data collection, refinement and validation statistics**

| Ligation mimic | Yes | Yes | Yes | No | | No | |
|---|---|---|---|---|---|---|---|
| UBE2R2 | Yes | Yes | Yes | No | | No | |
| NEDD8 | Yes | Yes | Yes | No | | Yes | |
| CRL | CRL2$^{FEM1C}$ (EMD-17803) (EMD-17822) (PDB 8PQL) | CRL2$^{VHL-MZ1}$ (EMD-18767) | CRL1$^{FBXW7}$ (EMD-17802) | CRL2$^{FEM1C}$ Map1 Map2 (Map1: EMD-17798) (Map2: EMD-17799) | | CRL2$^{FEM1C}$ Map1 Map2 (Map1: EMD-17800) (Map2: EMD-17801) | |
| **Data collection and processing** | | | | | | | |
| Microscope | Krios | Krios | Glacios | Glacios | | Glacios | |
| Magnification | 130,000 | 130,000 | 22,000 | 22,000 | | 22,000 | |
| Voltage (kV) | 300 | 300 | 200 | 200 | | 200 | |
| Electron exposure (e–/Å$^2$) | 66 | 66 | 59 | 60 | | 60 | |
| Defocus range (µm) | −0.6 ~ −2.2 | -0.6 ~ -2.2 | −0.6 ~ −2.6 | −0.6 ~ −2.6 | | −0.6 ~ −2.6 | |
| Pixel size (Å) | 0.8512 | 0.8512 | 1.885 | 1.885 | | 1.885 | |
| Symmetry imposed | C1 | C1 | C1 | C1 | | C1 | |
| Initial particle images (no.) | 4,187,858 | 1,210,530 | 2,989,541 | 2,801,309 | | 2,963,610 | |
| Final particle images (no.) | 61,956* | 12,520 | 65,467 | 56,038 | 38,547 | 51,322 | 28,677 |
| Map resolution (Å) FSC threshold | 3,76** (0,143) | 7,5 (0,143) | 8,1 (0,143) | 7,72 (0,143) | 7,54 (0,143) | 7,19 (0,143) | 6,88 (0,143) |
| Map resolution range (Å) | – | – | – | – | – | – | – |
| **Refinement** | | | | | | | |
| Initial model used (PDB code) | 5N4W 6NYO 6TTU 6LBN | | | | | | |
| Model resolution (Å) FSC threshold | 3.8 (0.143) | | | | | | |
| Model resolution range (Å) | | | | | | | |
| Map sharpening B factor (Å$^2$) | −70*** | −160 | −350 | −250 | −300 | −300 | −150 |
| Model composition | | | | | | | |
| Non-hydrogen atoms | 13554 | | | | | | |
| Protein residues | 1837 | | | | | | |
| Ligands | 3(ZN) | | | | | | |
| B factors (Å$^2$) | | | | | | | |
| Protein | 77.63 | | | | | | |
| Ligand | 92.01 | | | | | | |
| R.m.s. deviations | | | | | | | |
| Bond lengths (Å) | 0.004 | | | | | | |
| Bond angles (°) | 0.729 | | | | | | |
| Validation | | | | | | | |
| MolProbity score | 1.96 | | | | | | |
| Clashscore | 7.85 | | | | | | |
| Poor rotamers (%) | 0.38 | | | | | | |
| Ramachandran plot | | | | | | | |
| Favored (%) | 90.54 | | | | | | |
| Allowed (%) | 9.46 | | | | | | |
| Disallowed (%) | 0.00 | | | | | | |

*Consensus map: 61,956; focused map 1: 61,956; focused map 2,3: 52,377; focused map 4,5: 55,024 particles **Consensus map: Consensus map: 3.76Å; focused map 1: 3.55Å; focused map 2: 3.76Å; focused map 3: 3.89Å; focused map 4: 3.89Å; focused map 5: 3.84Å ***Consensus map: -70; focused map 1: -60; focused map 2: -80; focused map 3: -50; focused map 4: -70; focused map 5: -60

domain is consistent with our finding that mutations at the interface resulted in increased $K_m$ values of unanchored acceptor ubiquitin for the UBE2R2-mediated poly-ubiquitylation complex (Fig. 3b and Table 2). The ultimate test of the role of the structurally observed interface would be if compensatory UBE2R2 and acceptor ubiquitin mutations rescue the interaction. Indeed, assaying two such sets of mutant combinations restored $K_m$ to near WT values (Fig. 3b, Extended Data Fig. 4b and Table 2). We surmise that this distinctive placement

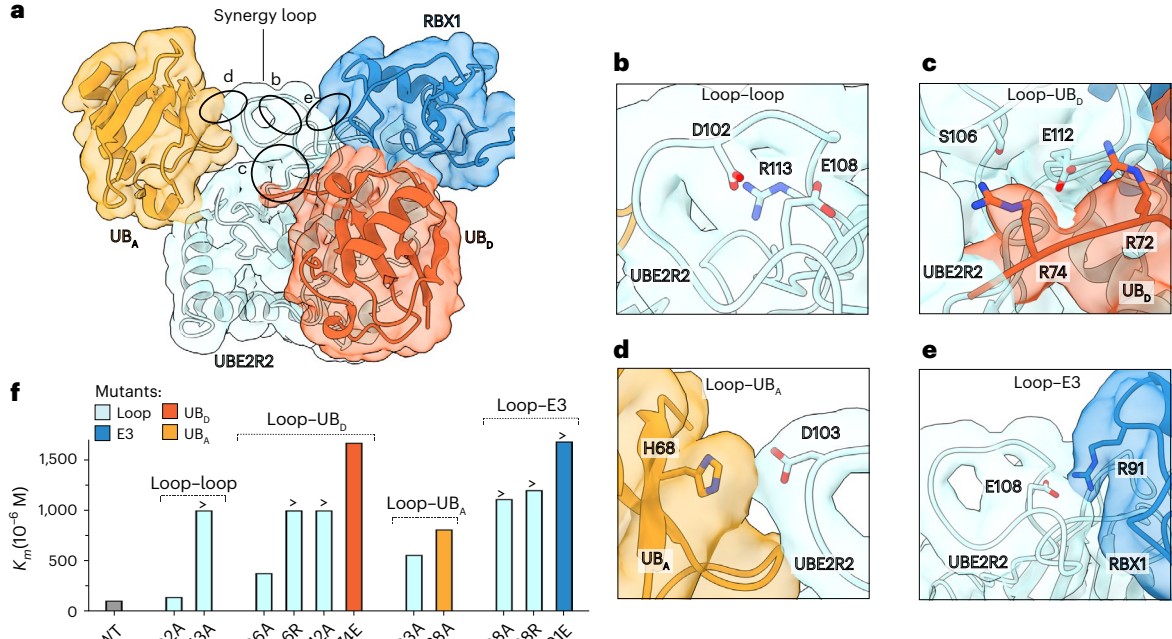

**Fig. 2 | CRL shaping of UBE2R2's synergy loop promotes millisecond poly-ubiquitylation by stabilizing donor and acceptor ubiquitins. a**, Ribbon diagram and cryo-EM density of the catalytic core, highlighting stabilizing interfaces between UBE2R2's synergy loop and the CRL subunit RBX1 (**e**), donor ubiquitin (UB_D) (**c**) and acceptor ubiquitin (UB_A) (**d**). **b**, Catalytic core cryo-EM density showing a close-up view of Asp102, Glu108 and Arg113 within the interaction hub. Subunits have been colored as in **a**. **c**, Same as in **b**, but showing proximity between residues Ser106 and Glu112 in the loop and Arg74 and Arg72 in

the donor ubiquitin, respectively. **d**, Same as in **b**, but highlighting the interaction between residue Asp103 in the loop and His68 in the acceptor ubiquitin. **e**, Same as in **b**, but showing the interaction between residue Glu108 in the synergy loop and Arg91 in RBX1. **f**, Bar graph comparing the $K_m$ values of unanchored acceptor ubiquitin for UBE2R2 and with WT or mutant proteins. Bars showing a '>' reflect reactions in which saturation of UBE2R2 with acceptor ubiquitin was not possible (the top concentration in the dilution series is shown). The value of each bar represents the estimated value for $K_m$ based on $n = 3$ technical replicates.

of the acceptor ubiquitin enables its dual engagement by the synergy loop to coordinate the catalytic assembly.

Overall, the structure showed multiple interactions contributing to the recruitment of donor and acceptor ubiquitins and poly-ubiquitylation of CRL substrates. In addition to contacts between UBE2R2 and the acceptor ubiquitin, CRL substrates are anchored by high-affinity interactions between their degrons and cognate receptors; here, the C terminus of the Sil1 peptide and FEM1C. To determine how the various mutant proteins would affect poly-ubiquitin chain formation onto a CRL-bound substrate, a method was devised to obtain highly pure mono-ubiquitylated Sil1 peptide (Extended Data Fig. 5a–c). This enabled our quantifying the rate of ubiquitin transfer ($k_{obs}$) in pre-steady state reactions performed on a quench flow instrument (Fig. 3c and Extended Data Fig. 5d). Interestingly, monitoring chain extension on ubiquitin-primed Sil1 peptide suggested that the high effective concentration afforded by degron binding largely masks defects caused by mutations in individual interaction surfaces (Fig. 3d, Extended Data Fig. 5e,f and Table 2). However, the roles of all the interfaces become apparent upon simultaneous mutation of any two. For example, mutation of the interface between the synergy loop and the CRL combined with either that between the acceptor ubiquitin and E2 UBC domain or the donor ubiquitin and the synergy loop resulted in 51-fold and 12-fold reductions in $k_{obs}$, respectively (Fig. 3d and Table 2). Similar effects were observed when the assays were repeated with UBE2R1 (Extended Data Fig. 5f,g and Table 2).

### General mechanism of poly-ubiquitin formation across CRLs

UBE2R2 mediates poly-ubiquitylation with numerous distinct CRLs. Previous structures have shown several different combinations of cullins, substrate receptors and substrates. We modeled their potential catalytic assemblies by docking them with our structure of RBX1, UBE2R2 and donor and acceptor ubiquitins. The models suggested that a common catalytic

architecture juxtaposes the poly-ubiquitylation active site and CRL-bound substrates (Extended Data Fig. 6a). To test our hypothesis of a common catalytic architecture, we applied cryo-EM to visualize UBE2R-mediated Lys48-linked chain extension for a ubiquitin-primed cyclin E phospho-peptide substrate of CRL1^FBXW7 (containing neddylated CUL1–RBX1 and substrate receptor SKP1–FBXW7; Tables 1 and 3). This is the human homolog of the archetypal CRL in yeast shown to work with Cdc34 (refs. 8–10). The map readily fit our coordinates for the catalytic assembly mediating poly-ubiquitylation (Fig. 4a and Supplementary Video 4).

To validate the structural data indicating that millisecond poly-ubiquitylation is a broad feature of CRLs, we generated additional ubiquitin-primed peptide substrates of various CRLs to estimate the pre-steady state kinetics of poly-ubiquitylation. The fastest rate of ubiquitin transfer was an astonishing ~100 s⁻¹ between ubiquitylated Hif1α peptide and neddylated CRL2 with VHL substrate receptor (Fig. 4b and Extended Data Table 1). The efficiency of product formation is striking, with 12% of Hif1α-ubiquitin further ubiquitylated in 2.5 ms, the limiting time of resolution for the quench flow instrument (Extended Data Fig. 6b). Overall, the average rate was 57 s⁻¹ for four distinct CRL complexes. Importantly, poly-ubiquitylation was again substantially slowed by mutations at the structurally observed interfaces (Fig. 4c, Extended Data Fig. 6c and Extended Data Table 1).

In addition to their endogenous targets, CRLs can serve as E3s, promoting targeted protein degradation in response to hetero-bifunctional molecules, termed proteolysis targeting chimeras (PROTACs)[53,54]. These agents trigger the ubiquitylation of a neo-substrate by inducing its proximity with an E3 (refs. 55–58). Several PROTACs use CUL2 and its substrate receptor VHL[59,60], with at least one such drug being explored as a cancer therapeutic in human clinical trials[61]. To determine whether the mechanism proposed here generalizes to neo-substrate poly-ubiquitylation, cryo-EM was performed on a neddylated CRL2^VHL complex in the presence of the PROTAC MZ1 (refs. 62,63)

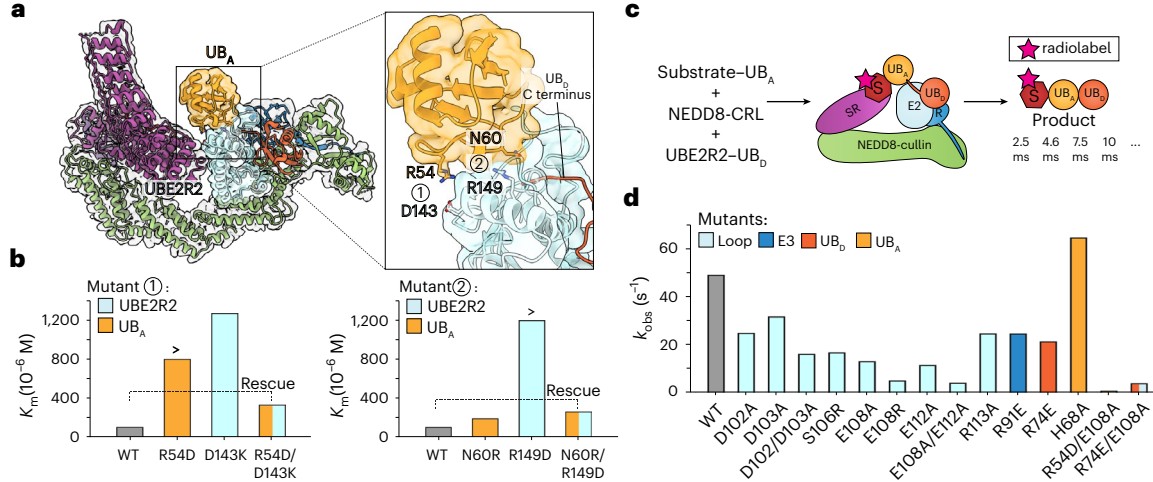

**Fig. 3 | UBE2R2's UBC domain assists the synergy loop to recruit the acceptor ubiquitin. a**, Ribbon diagram of the chain formation complex highlighting the UBE2R2–acceptor ubiquitin (UB$_A$) interface and the positions of key residues and their side chains mediating the interaction. Cryo-EM density from the composite map is shown. **b**, Bar graphs comparing the $K_m$ values for the indicated proteins of unanchored UB$_A$ for UBE2R2 in the presence of neddylated CUL2–RBX1. Compensatory mutations in UBE2R2 and UB$_A$ re-establish the interface, reducing $K_m$ values to near WT. Bars showing a '>' reflect reactions in which saturation of UBE2R2 with UB$_A$ was not possible (the top concentration in the dilution series

is shown). The value of each bar represents the estimated value for $K_m$ based on $n$ = 3 technical replicates. **c**, Schematic showing the assay used to estimate the rates of chain extension onto ubiquitin-primed CRL substrates by UBE2R-family E2s (also see Extended Data Fig. 5d for an illustration of quench flow operation). SR, substrate receptor; S, substrate; R, RBX1, UB$_D$, donor ubiquitin. **d**, Bar graph comparing the rates ($k_{obs}$) of donor ubiquitin transfer from UBE2R2 to ubiquitin-primed Sil1 peptide substrate for the indicated proteins. The value of each bar represents the estimated value of $k_{obs}$ based on $n$ = 3 technical replicates.

and a ubiquitin-primed BRD4 fragment that had been chemically linked to UBE2R2-ubiquitin (Table 1). The electron density maps were sufficiently resolved to yield the following conclusions: (1) the catalytic core, including UBE2R2-donor ubiquitin, acceptor ubiquitin and the RBX1 RING domain, readily fit into the electron density (Fig. 4d and Supplementary Video 4); (2) the broad functionality of the UBE2R2 synergy loop also appeared to be conserved, including its interactions with the RING and both donor and acceptor ubiquitins (Fig. 4d inset); (3) the CRL complex accommodated the catalytic core by subtle rearrangement of the RING domain (Extended Data Fig. 6d), presumably due to the larger neo-substrate compared to peptides; (4) mutations in residues within the UBE2R2 UBC domain and located at the interface with acceptor ubiquitin led to defects in neo-substrate poly-ubiquitylation (Fig. 4e and Extended Data Fig. 6e, f); and (5) both the WHB domain and NEDD8 appeared to be disordered owing to a lack of clear electron density.

### Comparison of CRL-mediated poly-ubiquitylation with RING E3s

To date, only two structures have been elucidated of RING-based E3s catalyzing Lys48-linked poly-ubiquitin chain formation. Superposition of the human E2 UBE2K and an associated acceptor ubiquitin[64] with equivalent molecular counterparts from the neddylated CRL2$^{FEM1C}$-based chain elongation structure showed distinct orientations of the acceptor ubiquitins relative to the E2 (Fig. 5a). Furthermore, the mechanism of poly-ubiquitin chain formation by the yeast E3 Ubr1 (ref. 65) also differed. First, this comparison showed the acceptor ubiquitins likewise interacting with their respective E2s through distinct conformations (Fig. 5b). Second, and in contrast with UBE2R2's synergy loop, Ubr1 contains a short stretch of residues that were disordered during substrate priming but appeared to stabilize the conformation of the acceptor ubiquitin during poly-ubiquitin chain formation (Fig. 5c–e). Consequently, the rate of Ubr1-catalyzed chain formation was slower than the substrate priming reaction[65]. Even greater differences were observed in the acceptor ubiquitin conformation when the neddylated CRL2$^{FEM1C}$ structure was compared to an E2-ubiquitin–E3 complex promoting Lys63-linked poly-ubiquitin chain formation[66] (Fig. 5f).

### Ubiquitin chain formation depends on unique CRL remodeling

CRLs are activated by the covalent linkage of NEDD8 to a conserved cullin lysine residue[20]. NEDD8 has been shown to stimulate the catalytic efficiency of substrate priming[43], in some cases by several orders of magnitude[22]. We and others[32,39] also found that this property was paralleled by UBE2R2-mediated poly-ubiquitylation of ubiquitin-primed substrates whereby neddylation decreased the $K_m$ of UBE2R2 for the CRL complex while also increasing the rate of ubiquitin transfer, $k_{obs}$ (Extended Data Table 1). Previous structures of CUL1–RBX1-based CRLs had shown NEDD8 assisting RBX1 in recruiting the enzymes that mediate substrate priming[21,22]. Therefore, it was surprising that NEDD8 and its covalently linked cullin domain (the WHB domain) were not visible in the structures (Figs. 1c and 4a,d).

Interestingly, deleting the cullin WHB domain from CRL1$^{βTRCP2}$, CRL1$^{FBXW7}$, CRL2$^{VHL}$ and CRL2$^{FEM1C}$ E3s stimulated the kinetics of UBE2R2-mediated ubiquitin chain extension to an extent similar to or even exceeding the effect of neddylation (Fig. 6a, Extended Data Figs. 7a,b and Extended Data Table 1). Importantly, this effect is specific to UBE2R2: activity with other RBX1 partner ubiquitin-carrying enzymes (UBE2D3 and ARIH1) decreased upon deleting either the CUL1 or CUL2 WHB domain, in accordance with NEDD8 mediating their recruitment (Extended Data Fig. 7c,d).

We gained further insights from cryo-EM data for CRL2 complexes without UBE2R2. The cryo-EM density maps for neddylated and unneddylated CRL2$^{FEM1C}$ were reminiscent of a CRL1 complex in that they resulted in multiple classes with distinct conformations[22]. In the previous study, the positions of CUL1's WHB and RBX1's RING domain, and NEDD8 when present, could not be assigned in any class because of poor density. Similarly, in cryo-EM maps of neddylated CRL2$^{FEM1C}$, neither CUL2's WHB domain nor NEDD8 could be unambiguously assigned (Extended Data Figs. 7e and 8a). However, one class for the unneddylated CRL2$^{FEM1C}$ showed CUL2's WHB domain roughly positioned as in the prior crystal structure[45], restraining RBX1's RING domain (Extended Data Figs. 7f,g and 8b). Modeling NEDD8 on this structure, based on a study suggesting that NEDD8 and its covalently linked cullin WHB domain adopt the same conformation for CUL1 and CUL2 (refs. 22,67), showed it clashing (Extended Data Fig. 7h). Thus,

## Table 2 | Estimates of $K_m$ and $k_{obs}$ for neddylated CRL2*-mediated poly-ubiquitin chain formation

| $UB_A$[a] | $UB_D$[b] | $UBE2R$[c] | RBX1 | $K_m$[d] (µM) | Fold change ($K_m$) | $k_{obs}$ S1-S2 (s⁻¹) | Fold change ($k_{obs}$) |
|---|---|---|---|---|---|---|---|
| **WT** | **K48R** | **WT (R2)** | **WT** | **111.33±7.58** | – | **45.55±2.88** | – |
| **WT** | **K48R** | **WT (R1)** | **WT** | **310.77±43.37** | – | **33.60±1.68** | – |
| **Interface 1: synergy loop–$UB_A$** | | | | | | | |
| H68A | K48R | WT (R2) | WT | 815.48±70.88 | 7.3 | 64.37±2.71 | 0.71 |
| R54D | K48R | WT (R2) | WT | >800 | >7 | 17.02±0.96 | 2.7 |
| N60R | K48R | WT (R2) | WT | 200.45±21.71 | 1.8 | – | – |
| WT | K48R | D143K (R2) | WT | 1,261.60±132.41 | 11.3 | 27.01±1.03 | 1.7 |
| WT | K48R | R149D (R2) | WT | >1,200 | >10 | 29.40±1.41 | 1.5 |
| R54D | K48R | D143K (R2) | WT | 334.51±59.99 | 3 | 31.68±1.49 | 1.4 |
| N60R | K48R | R149D (R2) | WT | 269.36±28.66 | 2.4 | 38.32±2.30 | 1.2 |
| WT | K48R | D103A (R2) | WT | 563.97±99.14 | 5.1 | 31.67±1.02 | 1.4 |
| **Interface 2: synergy loop–synergy loop** | | | | | | | |
| WT | K48R | D102A (R2) | WT | 147.66±8.03 | 1.3 | 24.94±1.35 | 1.8 |
| WT | K48R | R113A (R2) | WT | >1,000 | >9 | 24.71±3.24 | 1.8 |
| WT | K48R | R113A (R1) | WT | >1,200 | >4 | 22.15±0.87 | 1.5 |
| **Interface 3: synergy loop–$UB_D$** | | | | | | | |
| WT | K48R | S106A (R2) | WT | 384.62±48.04 | 3.5 | – | – |
| WT | K48R | S106R (R2) | WT | >1,000 | >9 | 16.84±1.40 | 2.7 |
| WT | K48R | E112A (R2) | WT | >1,000 | >9 | 11.61±1.01 | 3.9 |
| WT | K48R/R74E | WT (R2) | WT | 1,667.90±180.82 | 15.0 | 21.37±0.74 | 2.1 |
| WT | K48R/R74E | WT (R1) | WT | >1,200 | >4 | 22.91±1.91 | 1.5 |
| **Interface 4: synergy loop–RING domain** | | | | | | | |
| WT | K48R | E108A (R2) | WT | >1,000 | >9 | 13.18±1.50 | 3.5 |
| WT | K48R | E108A (R1) | WT | >1,200 | >4 | 15.29±0.66 | 2.2 |
| WT | K48R | E108R (R2) | WT | >1,200 | >10 | 5.17±0.21 | 8.8 |
| WT | K48R | WT (R2) | R91E | >1,680 | >15 | 24.67±1.59 | 1.8 |
| WT | K48R | WT (R1) | R91E | >1,200 | >4 | 8.92±0.62 | 3.8 |
| **Interface 5: multiple interfaces** | | | | | | | |
| WT | K48R | D102A/103A (R2) | WT | 924.45±126.37 | 8.3 | 16.21±0.73 | 2.8 |
| WT | K48R | E108/112A (R2) | WT | >1,200 | >10 | 4.23±0.14 | 10.8 |
| R54D | K48R | E108A (R2) | WT | >800 | >7 | 0.89±0.06 | 51.2 |
| R54D | K48R | E108A (R1) | WT | >800 | >2.5 | 0.68±0.06 | 49.4 |
| WT | K48R/R74E | E108A (R2) | WT | >1,200 | >10 | 3.68±0.33 | 12.4 |

*Experiments estimating $k_{obs}$ included Elongin B/C–FEM1C and Sil1-ubiquitin substrate; [a]Acceptor ubiquitin; refers to either unanchored ubiquitin for $K_m$ (D77 ubiquitin; see Methods) or conjugated to Sil1 peptide ($k_{obs}$); [b]Donor ubiquitin; [c]UBE2R1 and UBE2R2 paralogs are denoted as (R1) and (R2), respectively; [d]$K_m$ of unanchored acceptor ubiquitin for UBE2R2, '>' denotes the highest concentration of the dilution series in cases for which $K_m$ could not be estimated; S1, Sil1-ubiquitin; S2, Sil1-ubiquitin₂. The standard error of measurements are shown for all estimates.

## Table 3 | Peptides

| Peptide | Sequence | Source |
|---|---|---|
| Cyclin E 'sortasing' assays | GGGGPLPAGLL(pT)PPQ(pS)GRRASY | [21] |
| Cyclin E 'sortasing' cryo-EM | GGGGLPSGLL(pT)PPQ(pS)GKKQSSDYKDDDDK | [21] |
| Cyclin E substrate assays | Ac-KAMLSEQNRASPLPSGLL(pT)PPQ(pS)GRRASY | [21] |
| β-catenin 'sortasing' assays | GGGGYLD(pS)GIH(pS)GATTAPRRASY | [22] |
| Hif1α 'sortasing' assays | GGGGLLA(hyP)PAAGDTIISLDFGSNGRRASY | MPI |
| Hif1α substrate assays | Ac-KLRREPDALTLLA(hyP)AAGDTIISLDFGSN-Fluorescein | MPI |
| Sil1 substrate assays | Ac-GRRASYGSGSKEGYFQELLGSVNPTQGRAR | NEP |

All peptides were either purchased from Vivitide (formerly New England Peptides (NEP); greater than 95% purity) or synthesized in-house at the Max Planck Institute of Biochemistry (MPI) and solubilized in water. All single lysine peptide substrates had their N termini acetylated (Ac). Phosphodegrons are shown as pT (phospho Thr) or pS (phospho Ser), and the hydroxylated Pro degron in Hif1α peptides are shown as hyP. The Sil1 peptide substrate amino acid sequence was based on the clone 13 design[17] that had optimized affinity for FEM1C. All peptides that were substrates for ubiquitylation assays contained the 'RRASY' amino acidic sequence that enabled 32P-labeling by protein kinase A (New England Biolabs).

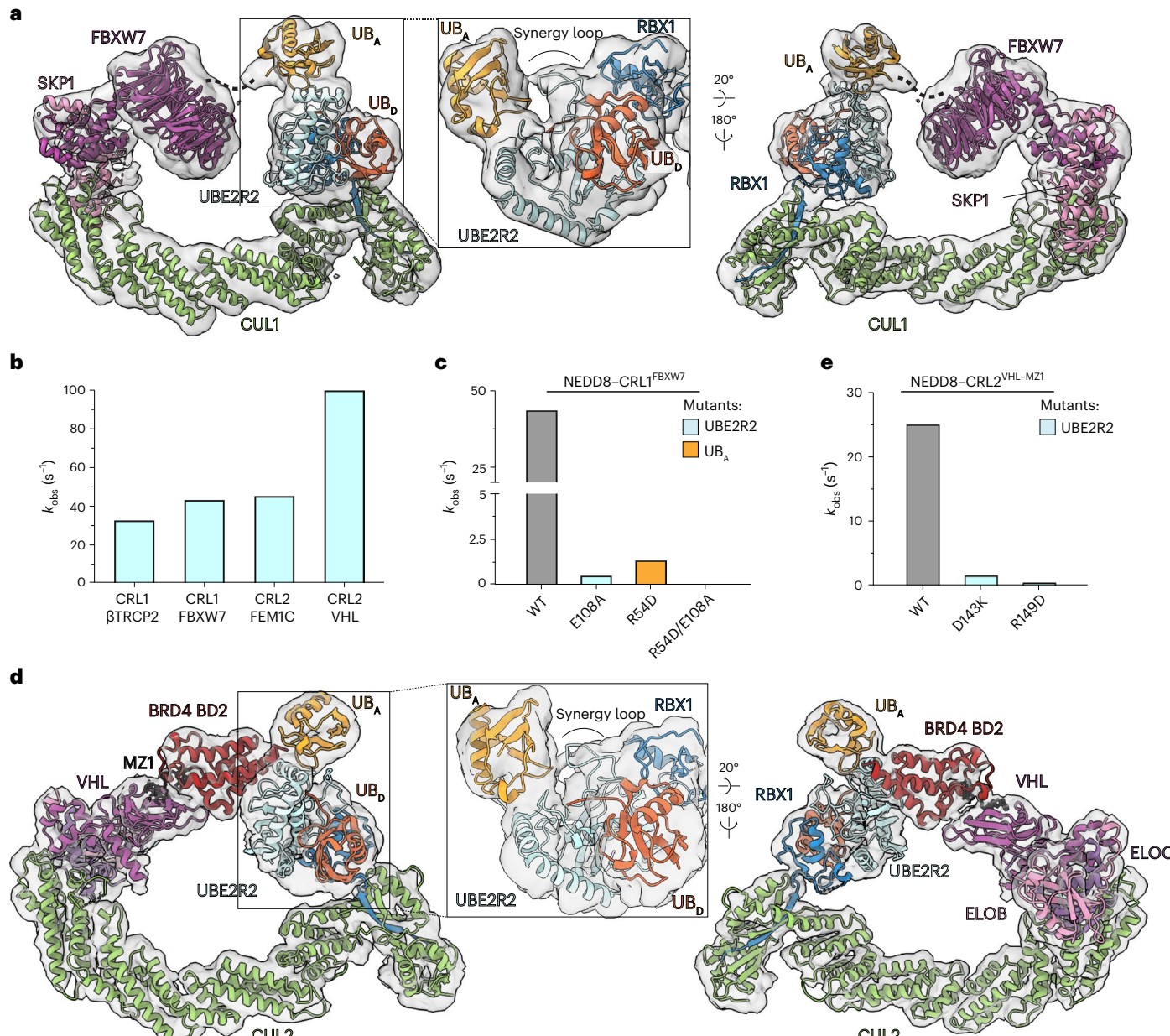

**Fig. 4 | A generalized catalytic core for UBE2R-mediated poly-ubiquitin chain formation. a**, Ribbon diagram of neddylated CUL1–RBX1 bound to SKP1–FBXW7 with trapped UBE2R2-ubiquitin covalently bonded to ubiquitin-primed cyclin E peptide substrate and corresponding cryo-EM density. SKP1–FBXW7 (PDB 2OVP), CUL1 (PDB 6TTU) and the catalytic core (UBE2R2-$UB_D$–$UB_A$–RBX1, this study) were fit into the density using rigid-body refinement (UCSF Chimera). **b**, Bar graph comparing the ubiquitin transfer rates ($k_{obs}$) for neddylated CRL1 and CRL2 complexes (substrate receptor identities are indicated on the *x* axis). The value of each bar represents the estimated value of $k_{obs}$ based on $n = 3$ technical replicates. **c**, Same as **b** except with cyclin E peptide–ubiquitin substrate and neddylated $CRL1^{FBXW7}$ in combination with the indicated proteins. **d**, Ribbon diagram of neddylated CUL2–RBX1 bound to Elongin B/C–VHL with trapped UBE2R2-ubiquitin covalently bonded to ubiquitin-primed BRD4 neo-substrate in the presence of the PROTAC MZ1 and corresponding cryo-EM density. Elongin B/C–VHL–MZ1 (PDB 5T35), CUL2 and the catalytic core (this study) were fit into the density as described above. **e**, Same as **c**, except with neddylated $CRL2^{VHL}$, ubiquitin-primed BRD4 neo-substrate (see Methods) and MZ1. $UB_D$, donor ubiquitin; $UB_A$, acceptor ubiquitin; ELOB, Elongin B; ELOC, Elongin C.

neddylation would trigger rearrangement of its covalently linked CUL2 domain and liberate interactions with the RBX1 RING.

We designed mutations to test whether the activating role of NEDD8 towards UBE2R2 is to remove the WHB domain from its perch. A NEDD8 Q40E mutant that would prevent interactions mediating its clashing with the RING domain greatly reduced the rate of poly-ubiquitylation with UBE2R2 (Extended Data Fig. 7i,j). Notably, Gln40 in NEDD8 is the primary target of pathogenic bacterial effector proteins that catalyze its deamidation, impairing CRL activity and ubiquitylation of their substrates[68,69]. Alternatively, charge-swapped point

mutants in CUL2 (D660K, E664K, D675K) were designed to liberate the WHB domain, activating UBE2R2-mediated ubiquitin chain extension in the absence of neddylation to an extent similar to neddylated WT CRL2 E3s (Fig. 6a,b, Extended Data Fig. 7a and Extended Data Table 1).

In summary, while CRL neddylation is required to promote both substrate priming and poly-ubiquitylation, NEDD8 does not directly mediate proximity between UBE2R2-ubiquitin and ubiquitin-primed substrate. Rather, it releases the RING domain from the cullin, which enables UBE2R2's unique interactions with both the RING and acceptor ubiquitin (Fig. 6c).

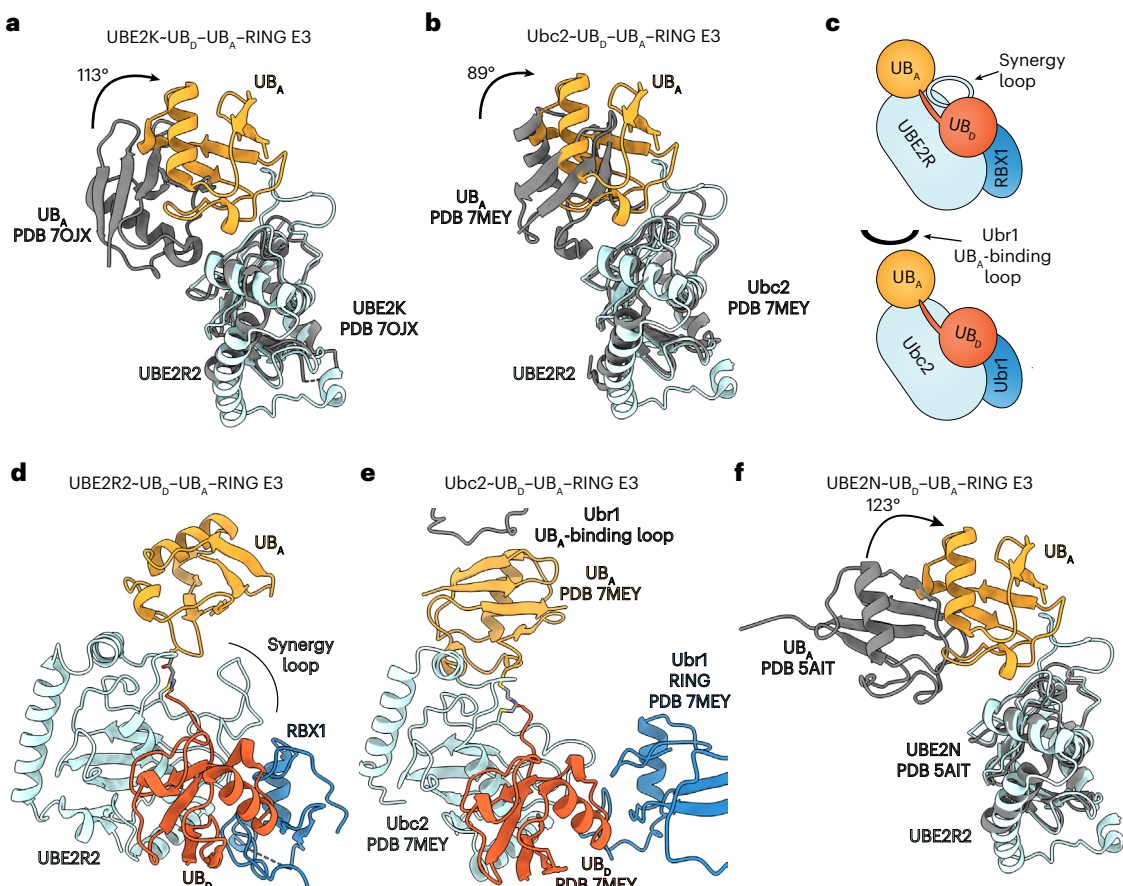

**Fig. 5 | Unique modes of acceptor ubiquitin activation during Lys48-linked poly-ubiquitin chain formation. a**, Structural superposition of UBE2R2 (cyan) and bound acceptor ubiquitin (UB$_A$; light orange) with the human E2 UBE2K and UB$_A$ (PDB 7OJX; gray). Notice dramatic rotation and translation of the acceptor ubiquitins relative to the E2 UBC domains. **b**, Same as **a**, except with the yeast E2 Ubc2 (PDB 7MEY). **c**, Cartoon illustrating distinct molecular solutions for controlling the orientation of UB$_A$ by UBE2R2's synergy loop (top) and the yeast E3 Ubr1's UB-binding loop (bottom). **d**, Ribbon diagram of the UBE2R2-UB$_D$-UB$_A$-RBX1 catalytic core (this study) during Lys48-dependent poly-ubiquitin chain formation. **e**, Same as **d**, except with the yeast E3 Ubr1 and Ubc2. **f**, Superposition of UBE2R2 (cyan) and UB$_A$ (light orange) with the human E2 UBE2N bound to UB$_A$ (gray) during Lys63-linked chain formation (PDB 5AIT). All structural alignments were performed with coordinates from the E2 UBC domains and UBE2R2–UB$_A$ from the neddylated CRL2$^{FEMIC}$ chain elongation structure. UB$_D$, donor ubiquitin.

## Discussion

Here, we show how CRLs together with Cdc34/UBE2R-family E2s rapidly forge the Lys48-linked ubiquitin chains that trigger timely degradation of their substrates. Numerous CRL-dependent interactions converge to adjoin the C terminus of the donor ubiquitin linked to UBE2R2's active site with Lys48 of a substrate-linked acceptor ubiquitin. Specifically, the CRL RING domain activates the catalytic conformation at multiple levels. In addition to the canonical function whereby the RING domain appears to directly stabilize the activated E2-donor ubiquitin conformation, the RING also configures the synergy loop to buttress the donor ubiquitin against UBE2R2. Consequently, this sculpting of the synergy loop also guides the acceptor ubiquitin. These interactions are interconnected; thus, the donor ubiquitin also shapes the synergy loop to organize the acceptor, and vice-versa (Fig. 2).

The multiplicity of interactions establishes extraordinary kinetics of poly-ubiquitin chain formation. Together with a CRL, UBE2R2 belongs to a small group of enzymes that are considered at or near catalytic perfection. The catalytic efficiency ($k_{cat}/K_m$) of such an enzyme is limited by the rate of forming the enzyme–substrate complex[70,71], ranging from $10^8$ to $10^9$ M$^{-1}$ s$^{-1}$. Here, the related value, $k_{obs}/K_m$, was estimated at nearly $10^8$ M$^{-1}$ s$^{-1}$ for neddylated CRL1$^{FBXW7}$ and a ubiquitin-primed cyclin E peptide substrate (Extended Data Table 1). Such rapid formation of Lys48-linked—that is degradative[72]—poly-ubiquitin chains would need to be tightly controlled to prevent wayward activity, perhaps explaining the requisite specificity of Cdc34/UBE2R-family E2s for CRLs.

In addition to a synergy loop, Cdc34/UBE2R-family E2s also contain a unique and conserved C-terminal extension. Like its synergy loop counterpart, the tail is also acidic and essential for viability in yeast[34,73,74]. Biochemically, the Cdc34/UBE2R-family C-terminal tail promotes processive poly-ubiquitylation, at least in part by enabling rapid rates of association between the E2 and the CRL complex through a basic canyon region on the cullin subunit[41]. Prior studies suggested that the acidic tail may adopt multiple conformations during cullin subunit binding[40], which seems consistent with an apparent lack of electron density for the various CRL-mediated chain elongation complexes reported here. The tail has also been shown to participate in catalysis[31], and our model for poly-ubiquitin chain formation suggests that tethering of the Cdc34/UBE2R UBC domain by its tail may restrain the conformational freedom of the active site towards the substrate, thus increasing the rate of ubiquitin transfer.

### CRLs as a paradigm for rapid poly-ubiquitylation

Our structural and quench flow kinetic data, taken together with previous knowledge of CRL regulation, suggest how rapid poly-ubiquitylation is tied to substrate binding to a cullin. Substrates effectively stimulate cullin neddylation by impeding NEDD8 deconjugation[75–79]. NEDD8, in turn, substantially potentiates all steps along the process of substrate poly-ubiquitylation. NEDD8 first stimulates the initial priming of substrates with ubiquitin. Previous structures showed the key role for NEDD8 as collaborating with CRLs to recruit

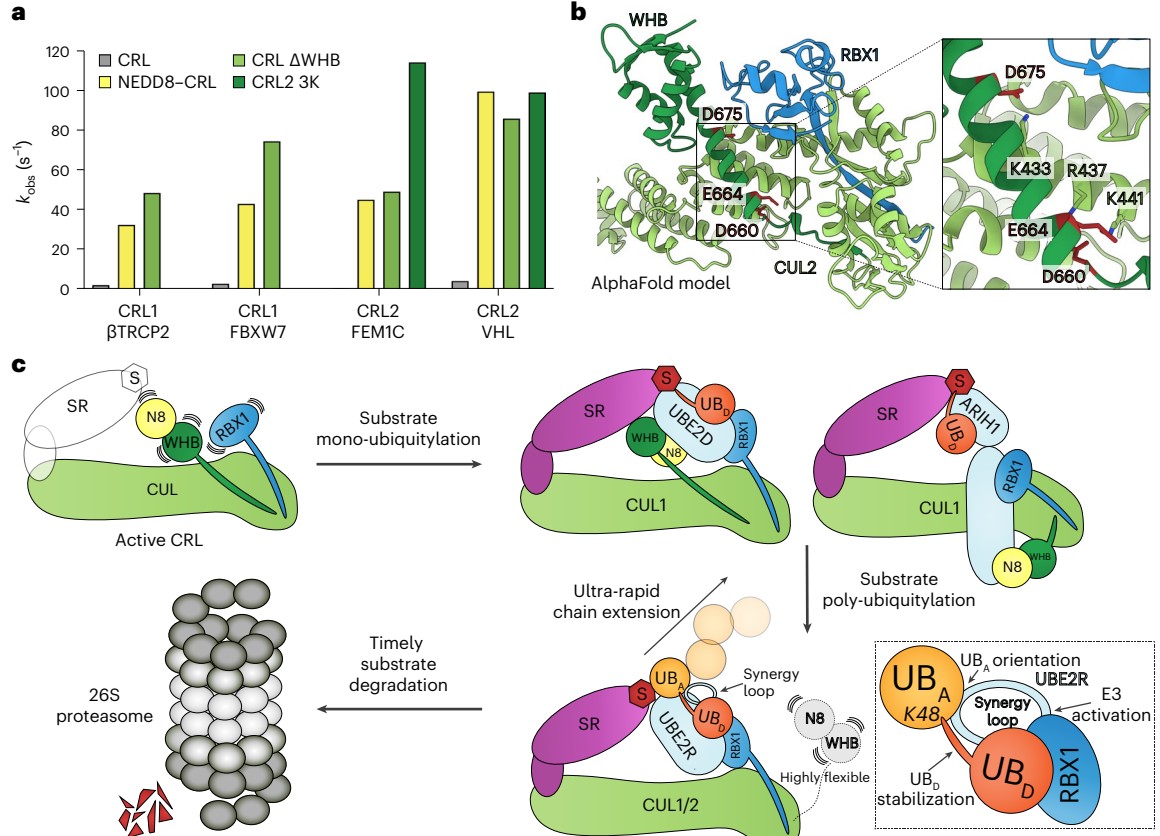

**Fig. 6 | Unique mechanisms guide CRL activation for substrate priming versus poly-ubiquitylation. a**, Bar graph showing $k_{obs}$ values for CRLs that were either unneddylated (gray), neddylated (yellow) or that contained mutant CUL1 or CUL2 subunits that lacked the WHB domain (ΔWHB; light green) or that disrupted interaction between the WHB domain and CUL2 (3K; dark green). The value of each bar represents the estimated value of $k_{obs}$ based on $n = 3$ technical replicates. **b**, Ribbon diagram of an AlphaFold model of the CUL2 C-terminal domain (green) bound to the RBX1 subunit (PDB 5N4W; blue). Charge-swapped mutations were introduced at three residues ('3K') in an α-helix buttressing the

CUL2 C/R domain, disrupting interaction between the WHB and the RBX1 RING domain. **c**, Model illustrating how CRLs achieve timely substrate degradation. CRLs are activated by NEDD8 (N8) conjugation to the cullin's WHB domain. Substrate priming is catalyzed by either the UBE2D-family E2s or the E3 ARIH1, with direct interaction between UBE2D and NEDD8 mediating juxtaposition between the donor ubiquitin and substrate. UBE2R-family E2s promote chain extension through their synergy loop and without direct interaction with NEDD8. $UB_D$, donor ubiquitin; $UB_A$, acceptor ubiquitin; SR, substrate receptor; S, substrate.

the enzymes mediating ubiquitin transfer directly to substrate[21,22,42]. We showed that NEDD8 also substantially activates the catalytic efficiency of ubiquitin chain extension on ubiquitin-primed substrates by nearly two orders of magnitude (Extended Data Table 1). Surprisingly, this does not involve direct interaction of NEDD8 with UBE2R2, but rather NEDD8-dependent release of a cullin's grip on the RBX1 RING domain.

Insertions in the same location as the Cdc34/UBE2R-family E2 synergy loop are found in only one other E2 family: Ubc7/UBE2G. Interestingly, UBE2G1 was recently shown to mediate targeted protein degradation relying on a CRL4 E3 through the formation of Lys48-linked poly-ubiquitin chains on neo-substrates[54,59,80]. In addition, UBE2G1 is necessary for the efficient degradation of CUL1-containing CRL substrates upon ablation of UBE2R1 and UBE2R2 (ref. 39). Furthermore, the structure of UBE2G2 bound to an E3 fragment showed interaction between the distal loop and the RING domain[81]. Therefore, it seems likely that the structural mechanism of UBE2G-mediated poly-ubiquitylation largely parallels that reported here for UBE2R2. We propose that the unique E2 synergy loop is a general strategy in nature to coordinate neddylated CRLs, their cognate E2-ubiquitin conjugates and placement of the acceptor ubiquitin to drive ultra-rapid poly-ubiquitylation, triggering degradation across thousands of CRL substrates (Fig. 6c).

Although currently there are only three atomic resolution structures of E2-RING E3 complexes catalyzing Lys48-linked poly-ubiquitin chain formation, two key observations were made based on their

comparison. First, although the E2 UBC domain structure is highly conserved, its interaction with acceptor ubiquitin to delicately place Lys48 into the active site shows apparent conformational variation amongst the different E2–E3 pathways (Fig. 5). Second, even though additional contacts with the acceptor ubiquitin appear to be important, the Cdc34/UBE2R synergy loop emanates from the E2, whereas the yeast E3 Ubr1 both anchors the acceptor ubiquitin while simultaneously binding to substrate. Based on these observations, it seems likely that additional mechanisms of Lys48-linked poly-ubiquitin chain formation await discovery. We anticipate that the structures presented here will serve as a basis for comparison, as we suspect many additional E2–E3 structures promoting poly-ubiquitylation will be described soon.

## Online content

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

## Methods

### Cloning, protein expression and purification

All constructs used in this study were made by using common molecular biology procedures. Protein sequence modifications were introduced using the QuikChange site-directed mutagenesis protocol (Agilent).

All proteins are of human origin. The UBE2R1 and UBE2R2 constructs (and all mutant derivatives) were expressed in the Rosetta (DE3) bacterial strain with either N-terminal glutathione-S-transferase (GST) or 6×His tags that were liberated during the purification process owing to a tobacco etch virus (TEV) protease cleavage site. GST–thrombin–VHL (residue 54 to C terminus) and GST–TEV–FEM1C were co-expressed with elongin B/C in BL21-Gold (DE3) bacterial cells. Proteins were purified by nickel-agarose or GST-sepharose affinity chromatography followed by overnight treatment with TEV or thrombin proteases. Next, select proteins were further purified by ion-exchange chromatography before size-exclusion chromatography into a buffer containing 25 mM HEPES pH 7.5, 150 mM NaCl and 1 mM DTT or storage buffer (30 mM Tris pH 7.5, 100 mM NaCl, 10% glycerol, 1 mM DTT). A 6×His-3C-K48C ubiquitin–Sil1 fusion construct was expressed in Rosetta (DE3) bacterial cells and purified by nickel-agarose affinity chromatography followed by size-exclusion chromatography into a buffer that contained 25 mM HEPES pH 7.5, 150 mM NaCl and 1 mM TCEP. The final amino acid sequence of K48C ubiquitin–Sil1 was:

N-term- HHHHHHSSGLEVLFQGPMQIFVKTLTGKTITLEVEPSDTIEN VKAKIQDKEGIPPDQQRLIFAGCQLEDGRTLSDYNIQKESTLHLVLRLRAAE GYFQELLGSVNPTQGRAR -C-term

A 2×Strep–TEV K48C ubiquitin–BRD4 (residues 352–460) C356A C357A C391A C429A fusion construct was expressed in Rosetta (DE3) bacterial cells and purified by StrepTactin-sepharose affinity chromatography followed by overnight TEV cleavage and size-exclusion chromatography into a buffer that contained 25 mM HEPES pH 7.5, 150 mM NaCl and 1 mM TCEP. The final amino acid sequence of K48C ubiquitin BRD4 (residues 352–460) was:

N-term- GSMQIFVKTLTGKTITLEVEPSDTIENVKAKIQDKEGIP PDQQRLIFAGCQLEDGRTLSDYNIQKESTLHLVLRLRAAGSGSGSE QLKAASGILKEMFAKKHAAYAWPFYKPVDVEALGLHDYADIIKHPMDM STIKSKLEAREYRDAQEFGADVRLMFSNAYKYNPPDHEVVAMARKLQD VFEMRFAKMPDE -C-term

Ubiquitins (including mutant derivatives) were expressed in BL21 (DE3) bacterial cells and purified through nickel-agarose affinity chromatography followed by size-exclusion chromatography into storage buffer consisting of 30 mM Tris pH 7.5, 100 mM NaCl, 10% glycerol and 1 mM DTT. Donor ubiquitins used in the unanchored, di-ubiquitin synthesis assays were expressed as GST–TEV–PKA–ubiquitin fusions and purified as previously described[82]. The PKA site contains the consensus phosphorylation sequence for protein kinase A (see below for the radiolabeling procedure). SKP1–FBXW7[Δdimerization] (residues 263 to C terminus), SKP1–βTRCP2, APPBP1–UBA3, UBE2M, NEDD8 and sortase A were purified as previously described[22,30,83–85]. All WT and mutant 2×Strep–Dac–TEV–CUL2 constructs, 2×Strep–Dac–TEV–CUL2 residues 1–660 (ΔWHB), His–MBP–TEV–RBX1 (5–C), CUL1, CUL1 residues 1–692 (ΔWHB), GST–TEV–RBX1 (5–C) and UBA1 constructs were generated in pLIB vectors[86] and used to form baculoviruses in Sf9 cells. Baculoviruses corresponding to CUL2 and His–MBP–TEV–RBX1 (5–C) or to CUL1 and GST–TEV–RBX1 (5–C) were used to transduce High Five (BTI-TN-5B1-4) cells and co-express the indicated protein subunits[22,83]. Proteins were purified by either StrepTactin or GST-sepharose affinity chromatography followed by overnight TEV cleavage. Next, proteins were subjected to ion-exchange chromatography followed by size-exclusion chromatography on a SuperDex 200 column that had been equilibrated in a buffer containing 25 mM HEPES pH 7.5, 150 mM NaCl and 1 mM DTT. WT CUL2 with an RBX1 subunit harboring an R91E mutation was purified as previously described[87] or through the method of baculoviruses as described above. All CUL1–RBX1 and CUL2–RBX1 complexes were covalently modified by NEDD8 (neddylated) as previously described[22,83,88]. Human BRD4 (residues 346–460) neo-substrate containing a C-terminal 'GRRASY' sequence was cloned using standard procedures and contained an N-terminal His-tag for purification by nickel affinity chromatography. The protein was first expressed in Rosetta (DE3) bacterial cells and purified by capture on nickel-agarose beads. Overnight TEV cleavage was followed by ion-exchange chromatography (HiTrap SP HP) and size-exclusion chromatography into a buffer that contained 25 mM HEPES pH 7.5, 150 mM NaCl and 1 mM DTT.

### Generation of ubiquitin-primed substrates

**Generation of ubiquitin-primed peptide substrates using 'sortasing'.** The sortase-mediated transpeptidation reaction was used to link ubiquitin[75]-SGSGSLPETGG-C-term to Hif1α, β-catenin and cyclin E peptides designed for sortasing (see Table 3 above)[21,22,30].

**Generation of ubiquitin-primed Sil1 peptide substrates.** To generate a ubiquitin-primed peptide substrate, one would ideally choose an E2 that is efficient at substrate priming but not poly-ubiquitin chain formation. In the literature, S138A UBE2R1 was shown to be less defective at substrate priming than poly-ubiquitin chain formation[38]. Those results inspired testing S138D UBE2R2, which was also motivated by the work on the E2 Ubc9 showing that residues at the structurally equivalent position can be either serine or aspartate[89]. S138D UBE2R2 was still defective at poly-ubiquitin chain formation, but quite active for substrate priming (Extended Data Fig. 5a). As such, unneddylated CUL2–RBX1 (1 μM), Elongin B/C–FEM1C (1 μM) and Sil1 peptide (100 μM) were first diluted in 1× reaction buffer (30 mM Tris-HCl pH 7.5, 100 mM NaCl, 5 mM MgCl₂, 2 mM DTT and 2 mM ATP). It was subsequently discovered that S138D UBE2R2 was not efficient at priming substrates for some mutant ubiquitins. In these cases, the E2 UBE2D3 was used instead (see below). In another tube of equal volume, human E1 (1 μM), 6×His-tagged ubiquitin (WT or mutant; 90 μM) and either S138D UBE2R2 (1 μM) or WT UBE2D3 (40 μM) were diluted in reaction buffer. For generation of Sil1-WT-ubiquitin, either S138D UBE2R2 or UBE2D3 were used for various preparations, whereas UBE2D3 was used for Sil1-R54D ubiquitin. Finally, S138D UBE2R2 was used for Sil1-H68A-ubiquitin and Sil1-N60R-ubiquitin. These solutions were pre-incubated for 15 min, then the E1 mix was combined with the E3 mix to initiate the ubiquitylation reaction and allowed to react overnight (>16 h) at room temperature (20–22 °C). The reaction was then diluted 1:10 in nickel-agarose binding buffer (30 mM Tris pH 7.5, 250 mM NaCl, 5% glycerol, 0.1% IgePal and 20 mM imidazole) and incubated with Ni-NTA agarose (Qiagen) for 1 h at 4 °C. The resin was then washed three times with nickel binding buffer before being eluted on a gravity column with 4 ml of nickel elution buffer (50 mM HEPES pH 8.0, 200 mM NaCl and 300 mM imidazole). The resulting eluate was first concentrated using an Amicon Ultra 4 ml centrifugal filter (Millipore-Sigma) and then injected onto a SuperDex 75 (GE Healthcare) column that had been equilibrated in storage buffer (30 mM Tris pH 7.5, 100 mM NaCl, 10% glycerol and 1 mM DTT) (Extended Data Fig. 5b,c). Fractions containing pure Sil1-ubiquitin protein were combined, concentrated and drop-frozen in liquid nitrogen for storage at −80 °C.

**Generation of ubiquitin-primed cyclin E peptide substrates for acceptor ubiquitin mutant analysis.** Generation of ubiquitin-primed cyclin E peptide substrates by enzyme-catalyzed conjugation of ubiquitin to an N-terminal lysine residue was performed similarly to the ubiquitin-primed Sil1 peptide substrates with the following modifications. An unneddylated SCF complex with SKP1–FBXW7 substrate receptor was used with the E2 UBE2D3 (2 μM) for both WT and R54D ubiquitin. The reactions were incubated at room temperature for 3 h before purification involving the same steps as described above for Sil1-ubiquitin.

## Peptide, donor ubiquitin and BRD4 neo-substrate $^{32}$P-labeling

All peptides were labeled at 10 μM or 50 μM final concentrations as follows. Peptides were diluted in 10× NEBuffer for Protein Kinases (New England Biolabs) followed by the addition of 40 μM γ$^{32}$P-labeled ATP (Perkin Elmer) and protein kinase A (2,500 units) and incubation at 32 °C for 2 h. Proteins and peptides labeled at 50 μM or above were first incubated with radio-labeled ATP for 1 h followed by the addition of unlabeled ATP (400 μM) for an additional 1 h. Donor ubiquitins were similarly labeled at a concentration of 200 μM, whereas BRD4 was labeled at 50 μM.

## Biochemical assays and enzyme kinetics

**Unanchored di-ubiquitin formation assay for UBE2R2 synergy loop alanine scan.** This assay is a modified version of a ubiquitylation reaction that had been used to estimate yeast Cdc34 unanchored chain formation[33]. Both the donor and acceptor ubiquitins contain mutations that restrict product formation to di-ubiquitin, greatly simplifying quantification of the reaction outcomes and interpretation of the results. Here, all radio-labeled donor ubiquitins contained the K48R mutation (hereafter K48R donor ubiquitin) to prevent its ability to act as an acceptor ubiquitin (given UBE2R's strong preference to build poly-ubiquitin chains with Lys48 specificity). Similarly, all acceptor ubiquitins contained an additional aspartate at the C terminus (hereafter D77 acceptor ubiquitin) that prevented thioester formation with the E2 and its acting as a donor ubiquitin. A mixture of 0.625 μM neddylated CUL2–RBX1 was prepared in reaction buffer (30 mM Tris-HCl pH 7.5, 100 mM NaCl, 5 mM MgCl$_2$, 2 mM DTT and 2 mM ATP). Then, 4 μl of the solution was aliquoted into Eppendorf tubes, followed by the addition of 1 μl of 500 μM D77 acceptor ubiquitin (WT or a mutant variant) to a final volume of 5 μl. In another tube, human E1 (1 μM) and WT or mutant UBE2R2 proteins (15 μM) were mixed with $^{32}$P-labeled K48R donor ubiquitin (24 μM) in reaction buffer. Both solutions were incubated at room temperature for 15 min. Reactions were initiated by mixing equal volumes from both tubes before being quenched in 2× SDS–PAGE buffer (100 mM Tris-HCl pH 6.8, 20% glycerol, 30 mM EDTA, 4% SDS and 4% β-mercaptoethanol) after a 15 s incubation period. For reactions with E108A or E112A UBE2R2, the incubation time was increased to 45 s. Donor ubiquitin substrate and di-ubiquitin product were separated by SDS–PAGE on 18% gels followed by autoradiography and detection on an Amersham Typhoon 5 imager (Cytiva). The intensities of substrate and product were quantified in ImageQuant software v.8.2.0.0 (Cytiva) and used to estimate the fraction of substrate that had been converted to product by dividing the signal of product by the total signal in the lane. See Extended Data Fig. 3e for a diagram of the assay.

**Comparison of $^{32}$P-labeled K48R ubiquitin with untagged K48R ubiquitin.** Di-ubiquitin synthesis reactions were performed to compare the activities of $^{32}$P-labeled K48R donor ubiquitin that contained an N-terminal amino acid sequence motif that enabled its phosphorylation and untagged K48R ubiquitin that had been purchased (LifeSensors). A tube was assembled with 0.5 μM E1, 24 μM K48R ubiquitin and 4 μM UBE2R2 diluted in reaction buffer (30 mM Tris-HCl pH 7.5, 100 mM NaCl, 5 mM MgCl$_2$, 2 mM DTT and 2 mM ATP). A separate tube was assembled containing 0.5 μM neddylated CUL2–RBX1 and 50 μM D77 acceptor ubiquitin (Extended Data Fig. 3e). Both tubes were then incubated at room temperature for 15 min. Reactions were initiated by mixing equal volumes from both tubes and then quenching them at various time points with 2× SDS–PAGE buffer (100 mM Tris-HCl pH 6.8, 20% glycerol, 30 mM EDTA, 4% SDS and 4% β-mercaptoethanol). Substrates and products were then separated on 18% Tris-Glycine gels by SDS–PAGE before being stained by Coomassie blue solution (20% methanol, 10% acetic acid, 0.1% Coomassie blue; Extended Data Fig. 3f).

**Estimation of the $K_m$ of unanchored acceptor ubiquitin for the UBE2R1- or UBE2R2–NEDD8–CUL2–RBX1 complex.** Di-ubiquitin synthesis reactions were prepared as follows. A twofold dilution series

of D77 acceptor ubiquitin was generated with 1× reaction buffer (30 mM Tris-HCl pH 7.5, 100 mM NaCl, 5 mM MgCl$_2$, 2 mM DTT and 2 mM ATP) in a volume of 4 μl and was aliquoted into autoclaved Eppendorf tubes. A separate dilution consisting of 2.5 μM neddylated CUL2–RBX1 without substrate receptors was also made with reaction buffer, and 1 μl of this solution was combined with the acceptor ubiquitin solutions leading to a final total volume of 5 μl. Subsequently, in another tube, 0.5 μM human E1, 24 μM $^{32}$P-labeled K48R donor ubiquitin and 15 μM UBE2R1 or UBE2R2 were diluted with 1× reaction buffer. For reactions that involved K48R/R74E donor ubiquitin, it was determined that an E1 concentration of 1 μM was necessary to saturate loading of the E2 (Extended Data Fig. 4e). These mixtures were then incubated for 15 min at room temperature. Reactions were initiated by mixing equal volumes of the E3-acceptor ubiquitin and UBE2R mixtures together. Reactions were quenched using 2× SDS–PAGE buffer (100 mM Tris-HCl pH 6.8, 20% glycerol, 30 mM EDTA, 1% bromophenol blue, 4% SDS and 4% β-mercaptoethanol) after a 10 s incubation period. Reactions containing E108A UBE2R1 and UBE2R2, E108R UBE2R2, and E112A UBE2R2 were incubated for 30 s. Reactions containing E108A/E112A UBE2R2, E108A UBE2R1 and UBE2R2 in combination with R54D acceptor ubiquitin, E108A UBE2R2 in combination with K48R/R74E donor ubiquitin, and WT UBE2R1 and UBE2R2 with K48R/R74E donor ubiquitin were incubated for 45 s. Substrates and products were resolved on 18% Tris-Glycine SDS–PAGE gels and were dried using a Hoefer Slab Gel Dryer GD 2000 before being exposed overnight on a phosphor screen. The screens were then imaged using an Amersham Typhoon 5 scanning imager (Cytiva). Quantification of substrates and products was performed using ImageQuant software v.8.2.0.0 (GE Healthcare), in which the fraction of di-ubiquitin product was estimated by dividing its signal by the total, including that for the substrate. The data were plotted and fit to the Michaelis–Menten model using nonlinear regression (GraphPad Prism 9).

**Estimating the CRL-dependent activation of UBE2R2 di-ubiquitin formation.** The di-ubiquitin synthesis assay was performed both in the absence or presence of neddylated CUL2–RBX1 in which the RBX1 subunit was WT or harbored an R91E mutant. For reactions containing a CRL, solutions were assembled in reaction buffer (30 mM Tris-HCl pH 7.5, 100 mM NaCl, 5 mM MgCl$_2$, 2 mM DTT and 2 mM ATP) containing 0.5 μM CRL and 50 μM D77 acceptor ubiquitin. Reactions performed in the absence of the CRL were also assembled in reaction buffer with 50 μM D77 acceptor ubiquitin. A separate solution containing 0.5 μM E1, 1 μM UBE2R2 and either 2 μM or 10 μM $^{32}$P-labeled K48R donor ubiquitin was diluted in reaction buffer (for reactions containing a CRL, 10 μM $^{32}$P-labeled donor ubiquitin was used). These solutions were then incubated at room temperature for 15 min. Reactions were initiated by mixing equal volumes from both solutions before being quenched in 2× SDS–PAGE buffer (100 mM Tris-HCl pH 6.8, 20% glycerol, 30 mM EDTA, 4% SDS and 4% β-mercaptoethanol) after the indicated incubation periods (see Extended Data Fig. 4c,d). Reaction products were separated by SDS–PAGE on 18% Tris-Glycine gels followed by autoradiography and detection on a Typhoon 5 image scanner. The products were quantified by ImageQuant software v.8.2.0.0, whereby the fraction of di-ubiquitin was estimated by dividing its signal by the total signal, including that for the substrate. The data were then normalized whereby the fraction of product values was multiplied by the ratio of $^{32}$P-labeled K48R donor ubiquitin and UBE2R2 levels. The data were fit to a linear model in GraphPad Prism using linear regression.

**Estimation of $k_{obs}$ from UBE2R1 or UBE2R2 to ubiquitin-primed Sil1 or Hif1α peptide substrates.** Ubiquitylation reactions were performed as single-encounter reactions between the ubiquitin-primed substrate and UBE2R2 as follows. A tube containing the CRL components was prepared with neddylated CUL2–RBX1, Elongin B/C–FEM1C or Elongin B/C–VHL and radio-labeled ubiquitin-primed substrate in reaction buffer (30 mM Tris-HCl pH 7.5, 100 mM NaCl, 5 mM MgCl$_2$, 2 mM DTT

and 2 mM ATP). In another tube, human E1, unlabeled K48R ubiquitin and UBE2R1 or UBE2R2 were diluted in reaction buffer. For reactions with Hif1α-ubiquitin, 20 μM unlabeled Hif1α substrate peptide (see Table 3 above) was added to the E2 mix to ensure single-encounter conditions between the radio-labeled substrate and the CRL[19]. For reactions with Sil1-ubiquitin, unlabeled peptide was unnecessary owing to the relatively slow off-rate of Sil1-ubiquitin from the CRL relative to the time course. For reactions involving unlabeled K48R/R74E donor ubiquitin, the human E1 concentration was doubled to ensure complete loading of the E2 before quench flow (Extended Data Fig. 4e). The CRL and E2 mixtures were incubated for 15 min at room temperature, followed by loading onto the left and right sample loops of a KinTek RQF-3 quench flow instrument. Timepoints were taken by combining the two mixtures with drive buffer (30 mM Tris–HCl pH 7.5 and 100 mM NaCl). Reactions were quenched in 5× SDS–PAGE buffer (250 mM Tris–HCl pH 6.8, 50% glycerol, 75 mM EDTA, 1% bromophenol blue, 10% SDS and 10% β-mercaptoethanol). Substrates and products were then resolved using 18% Tris-Glycine SDS–PAGE gels, which were subsequently dried using a Hoefer Slab Gel Dryer model GD 2000 before being exposed overnight on a phosphor screen. The screens were then imaged using a Typhoon 5 scanner. Quantification of substrates and products was performed using ImageQuant software v.8.2.0.0 (GE Healthcare), in which the fraction of remaining Sil1-ubiquitin or Hif1α-ubiquitin was estimated by dividing the signal of the substrate by the total, including the products. The data were fit to a one-phase decay equation using nonlinear regression (GraphPad Prism 9). Final concentrations of all reagents can be found in the table below. UBE2R1 and UBE2R2 concentrations were chosen to saturate the CRL substrate complex when feasible. A schematic of the experimental setup for this method can be found in Fig. 3c and Extended Data Fig. 5d.

| Final conditions for neddylated CRL2$^{VHL}$ and CRL2$^{FEM1C}$ ubiquitylation reactions | | | | | | |
|---|---|---|---|---|---|---|
| Substrate receptor | Experiment | [E1] (μM) | [UB] (μM) | [UBE2R1/2] (μM) | [CRL2] (μM) | [Substrate] (μM) |
| VHL | QF | 0.25 | 10 | 5 | 0.25 | 0.1 |
| FEM1C | QF | 0.25 | 2 | 1 | 0.25 | 0.1 |

QF, quench flow.

**Comparison of E1-catalyzed charging of K48R and K48R/R74E donor ubiquitins onto UBE2R2.** A tube with 16.6 μM UBE2R2 and 27 μM K48R or K48R/R74E ubiquitin was diluted in non-reducing reaction buffer (30 mM Tris–HCl pH 7.5, 100 mM NaCl, 5 mM MgCl$_2$ and 2 mM ATP); 9 μl of this solution was then aliquoted to autoclaved reaction tubes. A twofold dilution series was generated with E1 protein in non-reducing reaction buffer, with the highest concentration being 10 μM. Reactions were initiated by adding 1 μl of the E1 serial dilution to each tube, followed by brief vortexing. After a 15 min incubation period at room temperature, reactions were quenched in non-reducing 2× SDS–PAGE buffer (100 mM Tris–HCl pH 6.8, 20% glycerol, 30 mM EDTA and 4% SDS). Products were separated by SDS–PAGE on 18% Tris-Glycine gels and then stained with Coomassie blue solution (20% methanol, 10% acetic acid and 0.1% Coomassie blue). The experiment was then repeated using $^{32}$P-labeled donor ubiquitins before exposure to a phosphor screen. Substrates and products were visualized by scanning on an Amersham Typhoon 5 (Cytiva) and quantified using ImageQuant software v.8.2.0.0 (GE Healthcare). The fraction of product signal was determined by dividing the UBE2R2-donor ubiquitin signal by the total signal in the lane (Extended Data Fig. 4e). Experiments were performed in duplicate technical replicates.

**Estimation of $k_{obs}$ from UBE2R2 to ubiquitin-primed cyclin E or β-catenin peptide substrates.** Reactions were performed as described above for Sil1-ubiquitin and Hif1α-ubiquitin with the following

exceptions. The CRL tube contained neddylated CUL1–RBX1, unmodified CUL1–RBX1 or CUL1$^{ΔWHB}$–RBX1 with either SKP1–FBXW7 or SKP1–βTRCP2 substrate receptors. In the E1 mix, either unlabeled cyclin E or β-catenin substrates were used instead of Hif1α (20 μM). For reactions that involved E108A UBE2R2 and/or cyclin E-Ub$^{R54D}$, the data were fit to analytical closed-form equations[19] in Mathematica (v.13.1).

| Final conditions for neddylated CRL1 or CUL1$^{ΔWHB}$ ubiquitylation reactions | | | | | | |
|---|---|---|---|---|---|---|
| Substrate receptor | Experiment | [E1] (μM) | [UB] (μM) | [UBE2R2] (μM) | [CRL1] (μM) | [Substrate] (μM) |
| FBXW7 | QF | 0.5 | 20 | 10 | 0.5 | 0.1 |
| βTRCP2 | QF | 0.5 | 20 | 10 | 0.5 | 0.1 |
| **Final conditions for CRL1 ubiquitylation reactions (unneddylated)** | | | | | | |
| Substrate receptor | Experiment | [E1] (μM) | [UB] (μM) | [UBE2R2] (μM) | [CRL1] (μM) | [Substrate] (μM) |
| FBXW7 | QF | 0.5 | 40 | 20 | 0.5 | 0.1 |
| βTRCP2 | QF | 0.5 | 40 | 20 | 0.5 | 0.1 |

QF, quench flow.

**Estimation of $k_{obs}$ from UBE2R2 to BRD4 (346–460) neo-substrate in complex with neddylated CRL2$^{VHL}$ and the PROTAC MZ1.** Generation of BRD4 (346–460)-ubiquitin fusion substrates is technically challenging. However, one can estimate the rates of poly-ubiquitin chain formation by first monitoring substrate priming followed by the formation of a di-ubiquitin chain on substrate, as described in an earlier pioneering work[19]. MZ1 PROTAC was purchased as a lyophilized powder (MedChemExpress) and solubilized in 100% DMSO at a concentration of 1 mM. Reactions were assembled in two separate mixtures as follows. In one tube, 0.5 μM neddylated CUL2–RBX1, 0.5 μM Elongin B/C–VHL, 0.5 μM $^{32}$P-labeled BRD4 (346–460) and 4 μM MZ1 were diluted in reaction buffer. In another tube, 0.5 μM E1, 20 μM WT ubiquitin and 10 μM WT UBE2R2 or its mutant derivatives were also diluted in reaction buffer. These tubes were then incubated for 15 min before being loaded into separate sample loops on a KinTek RQF-3 Quench-Flow instrument. Reactions were initiated by bringing the two solutions together with drive buffer (30 mM Tris pH 7.5 and 100 mM NaCl) and then quenching at various time points with 2× SDS–PAGE buffer (100 mM Tris–HCl pH 6.8, 20% glycerol, 30 mM EDTA, 4% SDS and 4% β-mercaptoethanol). Substrates and products were then resolved using 18% Tris-Glycine SDS–PAGE gels that were dried and exposed overnight to a phosphor screen. Images were collected by scanning of the phosphor screens on an Amersham Typhoon 5 (Cytiva) and quantified using ImageQuant software v.8.2.0.0 (GE Healthcare). Note that the time resolution was sufficient to separate the appearance of primed substrate and poly-ubiquitin chain formation, thus enabling their quantification (Extended Data Fig. 6e,f) The depletion of unmodified BRD4 (346–460) was determined by dividing the signal of the substrate by the total signal in the lane including products. The levels of mono-ubiquitylated BRD4 (346–460) were also determined by dividing the signal of BRD4 (346–460)-ubiquitin by the total signal in the lane. The rates of ubiquitin transfer of priming and poly-ubiquitin chain extension were determined by fitting to their respective analytical closed-form solutions[19] in Mathematica (v.13.1).

**Multi-turnover reactions with ubiquitin-primed peptide substrates and estimation of the $K_m$ of UBE2R2 for various CRL1s.** All tubes were assembled in reaction buffer in a stepwise manner. First, the solution for tube one was prepared by the addition of CRL (neddylated CUL1–RBX1, CUL1–RBX1 or CUL1$^{ΔWHB}$–RBX1), substrate receptor complex (SKP1–βTRCP2 or SKP1–FBXW7) and followed by $^{32}$P-labeled peptide substrate (β-catenin-ubiquitin or cyclin E-ubiquitin). Next, the solution for tube two was assembled by the addition of E1 and unlabeled K48R donor ubiquitin, followed by a 1 min incubation period. Equal volumes

of the tube two solution were then aliquoted into nine autoclaved Eppendorf tubes. Subsequently, a twofold dilution series was formed for UBE2R2 and then sequentially added to the aliquoted tube two solutions which were incubated for 15 mins at room temperature. Initiation of the ubiquitylation reactions was accomplished by combining equal volumes of tube one and tube two solutions. All reactions were quenched with 2× SDS–PAGE buffer (100 mM Tris-HCl pH 6.8, 20% glycerol, 30 mM EDTA, 4% SDS and 4% β-mercaptoethanol) after a 10 s incubation period. Reactions were processed by SDS–PAGE, substrates and products were quantified (ImageQuant v.8.2.0.0) and the fraction of product formation was estimated. The fraction product was plotted as a function of the UBE2R2 concentration and fit to the Michaelis–Menten model using nonlinear regression (GraphPad Prism 9).

**Final conditions for neddylated CUL1–RBX1 and CUL1$^{\Delta WHB}$–RBX1 ubiquitylation reactions**

| Substrate receptor | Experiment | [E1] (µM) | [UB] (µM) | [UBE2R2] (µM) | [CRL1] (µM) | [Substrate] (µM) |
|---|---|---|---|---|---|---|
| FBXW7 | $K_m$ | 0.5 | 20 | 6.5* | 0.5 | 5 |
| βTRCP2 | $K_m$ | 0.5 | 20 | 6.5* | 0.5 | 5 |

*The top concentration in a twofold dilution series

**Final conditions for unmodified CUL1–RBX1 ubiquitylation reactions**

| Substrate receptor | Experiment | [E1] (µM) | [UB] (µM) | [UBE2R2] (µM) | [CRL1] (µM) | [Substrate] (µM) |
|---|---|---|---|---|---|---|
| FBXW7 | $K_m$ | 0.5 | 50 | 26* | 0.5 | 5 |
| βTRCP2 | $K_m$ | 0.5 | 50 | 26* | 0.5 | 5 |

*The top concentration in a twofold dilution series

**Steady state control reactions for the ΔWHB mutation in cullin.** All tubes were assembled in reaction buffer in a stepwise manner. First, tube one was prepared by the addition of CRL (neddylated CUL1–RBX1, CUL1–RBX1, CUL1$^{\Delta WHB}$–RBX1, neddylated CUL2–RBX1, CUL2–RBX1 or CUL2$^{\Delta WHB}$–RBX1), substrate receptor complex (SKP1–βTRCP2 or Elongin B/C–FEM1C) and then $^{32}$P-labeled substrate (β-catenin-ubiquitin or Sil1-ubiquitin). Next, tube two was prepared by the addition of E1 and WT donor ubiquitin followed by a 1 min incubation period and then UBE2D3, or ARIH1 with UBE2L3. For tube two mixtures containing UBE2D3, an additional 2 min incubation period was performed before initiation of the ubiquitylation reactions. For tube two mixtures containing ARIH1, the E2 UBE2L3 was added first and incubated for 2 min followed by the addition of ARIH1. All incubation periods were performed at room temperature. Reactions were initiated by combining equal volumes of tube one with tube two at room temperature. All reactions were quenched with 2× SDS–PAGE buffer (100 mM Tris-HCl pH 6.8, 20% glycerol, 30 mM EDTA, 4% SDS and 4% β-mercaptoethanol) at the indicated time points (Extended Data Fig. 7c,d). Reactions were processed by SDS–PAGE, substrates and products were quantified (ImageQuant v.8.2.0.0) and the fraction of product formation was estimated.

**Final conditions for steady state control reactions with UBE2D3**

| E3* | [E1] (µM) | [UB] (µM) | [UBE2D3] (µM) | [CRL] (µM) | [Substrate] (µM) |
|---|---|---|---|---|---|
| CRL2$^{FEM1C}$ | 0.5 | 30 | 12.5 | 0.25 | 0.1 |
| CRL1$^{BTRCP2}$ | 0.5 | 30 | 12.5 | 0.5 | 0.1 |

**Final conditions for steady state control reactions with ARIH1**

| E3* | [E1] (µM) | [UB] (µM) | [ARIH1] (µM) | [UBE2L3] (µM) | [CRL] (µM) | [Substrate] (µM) |
|---|---|---|---|---|---|---|
| CRL2$^{FEM1C}$ | 0.5 | 12.5 | 5 | 10 | 0.25 | 0.1 |

*E3 represents all three unneddylated, neddylated and ΔWHB CRL1 and CRL2 complexes

**Ubiquitylation assays comparing WT and S138D UBE2R2 substrate priming and poly-ubiquitin chain extension.** For this step, 1 µM WT or S138D UBE2R2 was charged with 20 µM WT ubiquitin by adding 1 µM E1 in reaction buffer (30 mM Tris pH 7.5, 100 mM NaCl, 5 mM MgCl$_2$, 2 mM ATP and 1 mM DTT) for 15 min at room temperature. Subsequently, the UBE2R2-WT ubiquitin solution was diluted twofold and incubated with 0.5 µM elongin B/C-VHL, 0.5 µM neddylated CUL2–RBX1 and 0.2 µM Hif1α peptide (that had been labeled with fluorescein) for the indicated time points (Extended Data Fig. 5a) before being quenched with 2× SDS–PAGE buffer (100 mM Tris-HCl pH 6.8, 20% glycerol, 30 mM EDTA, 4% SDS and 4% β-mercaptoethanol). Substrates and products were separated by SDS–PAGE and scanned on an Amersham Typhoon system (GE Healthcare).

### Generation and purification of activity-based probes

Activity-based probes (ABPs) were used to mimic the native intermediate of donor ubiquitin transfer to CRL substrate-linked acceptor ubiquitin by UBE2R2 (see Fig. 1b). The linear fusions of a Sil1 peptide that had been optimized for FEM1C binding[17] or BRD4 (352–460) were produced with K48C acceptor ubiquitin and expressed in bacteria (see both Table 3 for the Sil1 amino acid sequence and the 'Cloning, expression and purification' section at the beginning of the Methods for the full fusion sequences). Similarly, a linear fusion of cyclin E peptide with K48C ubiquitin was generated via the sortase-mediated transpeptidation reaction. All ABPs used His-tagged-ubiquitin(1–75)–MESNa and its conjugation to the compound (E)-3-[2-(bromomethyl)-1,3-dioxolan-2-yl]prop-2-en-1-amine (BmDPA) as previously described[21]. Reactive His-ubiquitin(1–75)–BmDPA (which mimics the donor ubiquitin in the final trapped complex; 0.5 mg ml$^{-1}$ final) was incubated with 100 µM K48C ubiquitin–Sil1 fusion, 100 µM K48C ubiquitin–BRD4 (352–460) or 100 µM K48C ubiquitin–cyclin E peptide for 1 h at 30 °C. The ABP was purified by size-exclusion chromatography in a column that had been equilibrated with a buffer containing 25 mM HEPES pH 7.5 and 150 mM NaCl.

### Formation of trapped CRL complexes

To form the trapped poly-ubiquitin chain-forming complex (containing neddylated CUL2–RBX1 and substrate receptor Elongin B/C–FEM1C, neddylated CUL2–RBX1 and substrate receptor Elongin B/C–VHL–MZ1 or neddylated CUL1–RBX1 and substrate receptor SKP1–FBXW7), UBE2R2 was incubated with 1 mM TCEP for 20 min on ice and desalted (Zeba spin columns) into a buffer that contained 25 mM HEPES pH 7.5 and 150 mM NaCl. Next, 7.5 µM desalted UBE2R2 was immediately added to other complex components including 7.5 µM neddylated CRL2$^{FEM1C}$, 7.5 µM neddylated CRL2$^{VHL-MZ1}$ or neddylated CRL1$^{FBXW7}$ and a sixfold molar excess of ABP. The reactions were incubated for 30 min at 30 °C. The trapped complex was purified by size-exclusion chromatography on a column that had been equilibrated in a buffer containing 25 mM HEPES pH 7.5, 75 mM NaCl and 1 mM TCEP.

### Cryo-EM

**Sample preparation.** The trapped poly-ubiquitin chain formation complexes were prepared as described in the previous section. The CRL2$^{FEM1C}$–Sil1 peptide and neddylated CRL2$^{FEM1C}$–Sil1 peptide complexes (that is, in the absence of E2 or donor ubiquitin) were formed by adding 7.5 µM CUL2–RBX1 or neddylated CUL2–RBX1, respectively, with 7.5 µM Elongin B/C–FEM1C substrate receptor complex and 7.5 µM Sil1 peptide substrate, and then incubated on ice for 15 min. Complexes were purified by size-exclusion chromatography on a column that had been equilibrated with a buffer containing 25 mM HEPES pH 7.5, 100 mM NaCl and 1 mM TCEP. A total of 3.5 µl of the purified complexes were applied onto R1.2/1.3 holey carbon grids (Quantifoil) and blotted with a blot force of 4 for 3.5 s using a Vitrobot Mark IV (4 °C, 100% humidity). Grids were subsequently plunge-frozen in liquid ethane.

**Data collection.** Three datasets (CRL2$^{FEM1C-Sil1}$, NEDD8–CRL2$^{FEM1C-Sil1}$ and NEDD8–CRL1$^{FBXW7-cyclin E-UB}$-UBE2R2-ubiquitin) were recorded on a 200 kV Glacios transmission electron microscope using a K2 direct detector set to counting mode and SerialEM software. The datasets consisted of 2,632, 2,820 and 3,192 micrographs, respectively, with a pixel size of 1.885 Å. The total exposure for each dataset was 59–60 electrons Å$^{-2}$ (40 frames), and the defocus value ranged between −0.6 and −2.6 µm.

The datasets for the NEDD8–CRL2$^{FEM1C-Sil1-UB}_A$–UBE2R2-ubiquitin and NEDD8–CRL2$^{VHL-MZ1-BRD4-UB}_A$–UBE2R2-ubiquitin complexes (14,022 and 4,848 total micrographs, respectively) were collected on a 300 kV Titan Krios transmission electron microscope with a pixel size of 0.851 Å, using a K3 direct detector in counting mode and SerialEM software. The total exposure was set to 66 electrons Å$^{-2}$ (38 frames) with a defocus ranging from −0.6 to −2.2 µm. Representative micrographs can be found in Supplementary Fig. 1.

**Data processing.** All datasets were processed in RELION v.3.1.1 (ref. 90). First, the raw movie frames were aligned and dose-weighted using MotionCorr2 v.1.1.0 (ref. 91). Second, CTFFIND4 was used to estimate the contrast-transfer function[92]. Particle picking was performed using Gautomatch v0.56. The ab initio NEDD8–CRL2$^{FEM1C-Sil1-UB}_A$–UBE2R2-ubiquitin trapped complex reconstruction was performed in cryoSPARC v4.2.0[93]. All further operations, including 2D classification, 3D classification, global, local and focused 3D refinement and post-processing, were done in RELION v.3.1.1. Processing schemes are shown in Extended Data Fig. 2 for the CRL2$^{FEM1C}$ chain formation structure and Extended Data Fig. 8 for both unmodified and neddylated CRL2$^{FEM1C}$ in the absence of UBE2R2 and donor ubiquitin. Cryo-EM data for the NEDD8–CRL1$^{FBXW7-cyclin E-UB}_A$–UBE2R2-ubiquitin and NEDD8–CRL2$^{VHL-MZ1-BRD4-UB}_A$–UBE2R2-ubiquitin chain formation complexes were processed in a similar manner as shown in Extended Data Fig. 2. For the latter complex, the map was kept binned (1.925 Å pixel size) owing to the reconstruction not reaching high resolution (the final resolution was 7 Å).

For the high-resolution, NEDD8–CRL2$^{FEM1C-Sil1-UB}_A$–UBE2R2-ubiquitin dataset, two classes were initially observed during data processing with differing density for the donor ubiquitin. As such, the final reconstruction was made using particles with visible donor ubiquitin. Additionally, to best extract certain features of the complex, a series of focused maps were generated, masking on: (1) CUL2–Elongin B/C–FEM1C$_{404-C}$–RBX1$_{5-35}$; (2) FEM1C$_{N-404}$–UBE2R2-UB$_D$–UB$_A$–Sil1–RBX1$_{35-104}$; (3) FEM1C$_{150-404}$–UBE2R2-UB$_D$–UB$_A$–Sil1–RBX1$_{35-104}$; (4) CUL2$_{429-556}$–FEM1C$_{N-404}$–UBE2R2-UB$_D$–UB$_A$–Sil1–RBX1$_{30-104}$; and (5) CUL2$_{429-556}$–FEM1C$_{150-404}$–UBE2R2-UB$_D$–UB$_A$–Sil1–RBX1$_{30-104}$ (Extended Data Fig. 2a). All final maps were post-processed using DeepEMhancer[94].

**Model building.** The high-resolution NEDD8–CRL2$^{FEM1C-Sil1-UB}_A$–UBE2R2-ubiquitin structure was built using five focused-refined maps, displaying high-resolution features of certain parts of the complex (Extended Data Fig. 2). Map one was used to build CUL2$_{1-563}$, Elongin B/C, FEM1C$_{404-C}$ and RBX1$_{21-35}$. FEM1C$_{150-404}$ was built using map two and Sil1 peptide substrate using map five. The UBE2R2 catalytic domain (except the synergy loop, residues 98–113) was built using map three while the synergy loop was modeled based on density from map four. The RBX1 RING domain (36–104) was built using maps three and five. Both donor and acceptor ubiquitins (UB$_D$ and UB$_A$) were built using maps three and five. The C/R domain of CUL2$_{563-644}$ and FEM1C's N-terminal domain (3–150) (PDB 5N4W and 6LBN, respectively) were docked into maps one and two, respectively, using rigid-body refinement in UCSF Chimera[95].

An initial model was made using previously determined crystal structures of single components or subcomplexes: UBE2R2 (PDB 6NYO), ubiquitin (donor and acceptor) and RBX1 (PDB 6TTU), CUL2–Elongin B/C (PDB 5N4W) and FEM1C$_{3-373}$ (PDB 6LBN). FEM1C$_{373-C}$ was built de novo using AlphaFold2 (ref. 96). The coordinates for these subunits were first manually placed into the cryo-EM maps followed by rigid-body refinement in UCSF Chimera[95]. Next, manual model building in COOT[97] and real-space refinement using Phenix.refine[98] were performed iteratively until good map-to-model correlations and geometries were achieved. Stretches of protein subunits that lacked clear electron density in the maps were excluded from the model. Side chains were built for residues that showed well-resolved density in the cryo-EM maps. For regions of the cryo-EM maps in which tracing of the backbone was feasible, coordinates for side chains from previous crystal structures were included or based on the results from protein mutagenesis performed in this study.

The composite map (Extended Data Fig. 2b) was constructed manually by combining better-resolved DeepEMhancer sharpened maps from focused refinements. Focused maps were also deposited as additional maps (EMD-17822). The consensus map served as a base for resampling of the focused maps. The composite map was used to build atomic coordinates of the structure. The final model for the neddylated CRL2$^{FEM1C-Sil1-UB}_A$–UBE2R2-ubiquitin complex was refined using the composite map. All figures displaying structures were generated with ChimeraX v1.4 software[99].

### Reporting summary

Further information on research design is available in the Nature Portfolio Reporting Summary linked to this article.

## Data availability

The atomic coordinates and electron microscopy maps have been deposited in the Protein Data Bank with accession code PDB 8PQL and in the Electron Microscopy Data Bank with codes EMD-17803 (consensus map) and EMD-17822 (composite map) for the neddylated CRL2$^{FEM1C}$ poly-ubiquitin chain formation complex, EMD-17802 for the neddylated CRL1$^{FBXW7}$ poly-ubiquitin chain formation complex, EMD-18767 for the neddylated CRL2$^{VHL-MZ1-BRD4}$ poly-ubiquitin chain formation complex, EMD-17798 and EMD-17799 for the CRL2$^{FEM1C}$ complex and EMD-17800 and EMD-17801 for the neddylated CRL2$^{FEM1C}$ complex. Publicly available PDB entries are 1LDJ, 1LM8, 4AP4, 5AIT, 5N4W, 6LBN, 6NYO, 6TTU, 7B5L, 7MEY and 7OJX. Source data are provided with this paper.

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

## Acknowledgements

We thank J. Kellerman, K. Baek, D. Horn-Ghetko, L. Hopf, L. Hehl, L. Henneberg, J. Basquin, S. von Gronau S. Uebel, S. Pettera and V. Sanchez for assistance, reagents and helpful discussions; and D. Bollschweiler, T. Schäfer and the cryo-EM facility at the Max Planck Institute of Biochemistry (MPIB). We also thank P. Laragan for technical assistance (University of Nevada, Las Vegas) and S. Übel and S. Petera in the MPIB Bioorganic Chemistry and Biophysics Core for peptide synthesis. Molecular graphics and analyses were performed with UCSF ChimeraX, developed by the Resource for Biocomputing, Visualization, and Informatics at the University of California, San Francisco, with support from National Institutes of Health (NIH) R01-GM129325 and the Office of Cyber Infrastructure and Computational Biology, National Institute of Allergy and Infectious Diseases. This research was supported by ALSAC, St. Jude Children's Research Hospital, NIH P30CA021765 to St. Jude, NIH R01GM125885 (D.C.S. and B.A.S.), NIH R01GM141409 (J. Li, N.P., C.R. and G.K.) and Max-Planck-Gesellschaft and the EU H2020 ERC Advanced Grant Nedd8Activate-789016 (J. Liwocha, S.M., D.T.K., J.R.P. and B.A.S.). The funders had no role in study design, data collection and analysis, decision to publish or preparation of the manuscript.

## Author contributions

J. Liwocha, J. Li, N.P., C.R., S.M., D.T.K., B.S. and D.C.S. generated protein complexes and/or performed biochemical and biophysical characterization. J. Liwocha designed and generated activity-based probes. J. Li, N.P., C.R. and G.K. performed ubiquitylation assays, kinetics experiments and analyzed the results. J. Liwocha and S.M. collected, processed and refined the neddylated CRL2^VHL-MZ1-BRD4 poly-ubiquitin chain formation complex dataset. J. Liwocha collected, processed and refined all other cryo-EM datasets. J. Liwocha and J.R.P. built and refined the poly-ubiquitin chain formation structure. J. Liwocha, B.A.S. and G.K. prepared the paper with input from all other authors. B.A.S. and G.K. supervised the project.

## Funding

## Competing interests

B.A.S. is a member of the scientific advisory boards of Proxygen and BioTheryX, and co-inventor of intellectual property licensed to Cinsano. The other authors declare no competing interests.

## Additional information

**Extended data** is available for this paper at https://doi.org/10.1038/s41594-023-01206-1.

**Correspondence and requests for materials** should be addressed to Brenda A. Schulman or Gary Kleiger.

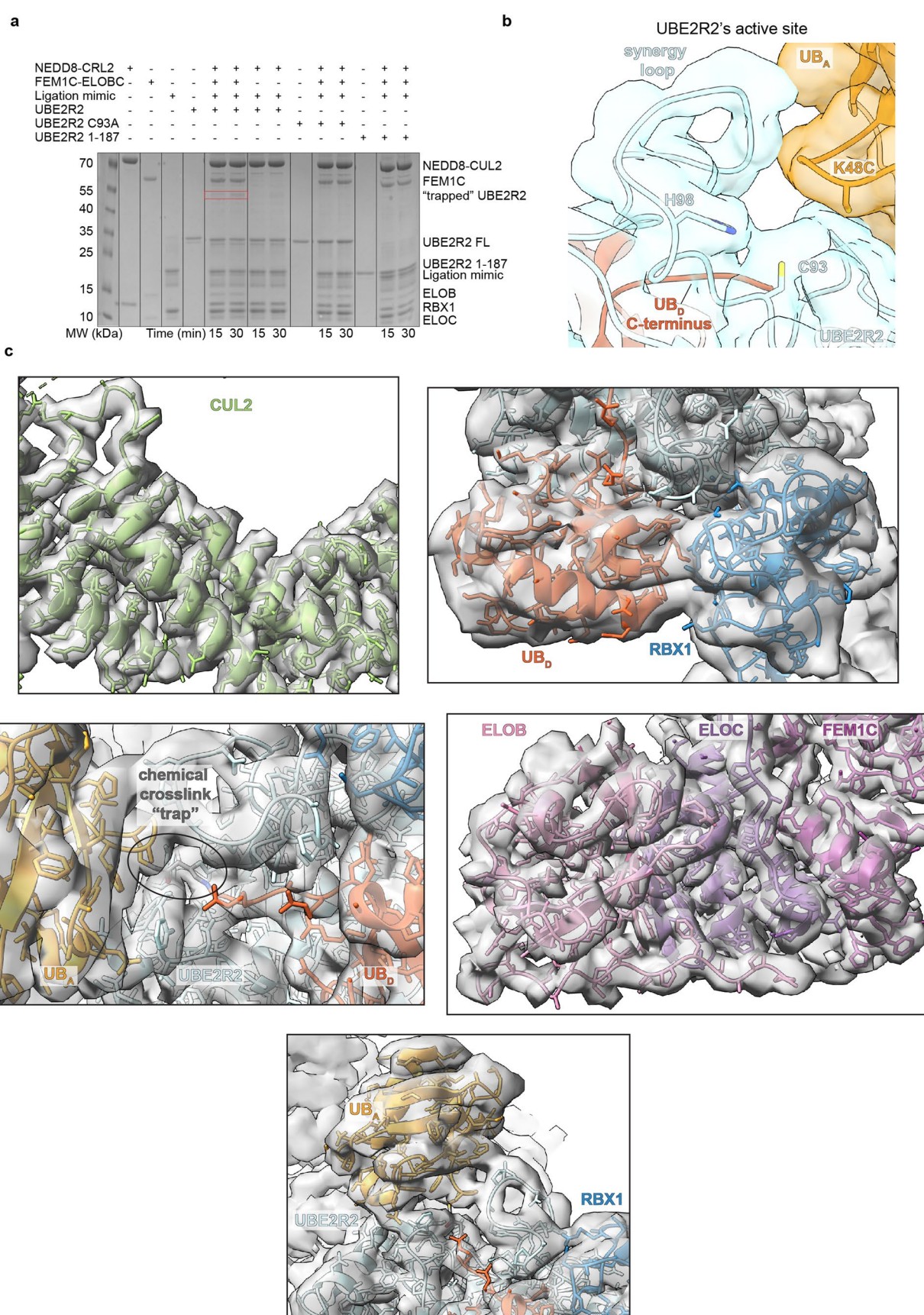

**Extended Data Fig. 1 | See next page for caption.**

**Extended Data Fig. 1 | Formation of an activity-based probe to capture millisecond poly-ubiquitin chain formation. a**, Coomassie-stained SDS-PAGE gel examining reactions containing wild-type (WT) or catalytically inactive UBE2R2 with the ligation mimic (defined as the donor ubiquitin cross-linked to the BmDPA molecule, see Methods) and in the absence or presence of neddylated CRL2$^{FEMIC}$ or separated CRL components. The red box identifies the "trapped" UBE2R2 - UB$_D$-substrate-UB$_A$ product. Notice that formation of the trapped complex was dependent on the presence of WT UBE2R2 containing its C-terminal acidic tail and an intact CRL containing the substrate receptor complex (FEM1C-ELOBC, Elongin B/C-FEM1C; UB$_D$, donor ubiquitin; UB$_A$, acceptor ubiquitin; UBE2R2 FL, full-length; MW, molecular weight). The gel is representative of duplicate technical replicates. **b**, Electron density from the composite cryo-EM map highlighting the UBE2R2 active site region including UB$_D$'s C-terminal tail, UB$_A$, and the UBE2R2 synergy loop. While visualization of UB$_A$'s Lys48 was not possible owing to its replacement with a Cys to promote trap formation, clear density is visible for synergy loop residue His98, previously shown to be important for UBE2R2 activity[32], whose side-chain imidazole ring points towards the UBE2R2 active site Cys93. **c**, Select regions of the composite cryo-EM map and ribbon diagrams of the corresponding protein subunits. Ball-and-sticks are shown for residues where side-chains were built. Top-left, the central stalk of CUL2; top-right, donor ubiquitin (UB$_D$) and RBX1; middle-left, the "trap" indicating the location for the ligation mimic-UBE2R2-crosslinked junction; middle-right, Elongin B/C with part of FEM1C; bottom, the catalytic core with UBE2R2, RBX1, and donor and acceptor ubiquitins (UB$_D$ and UB$_A$, respectively).

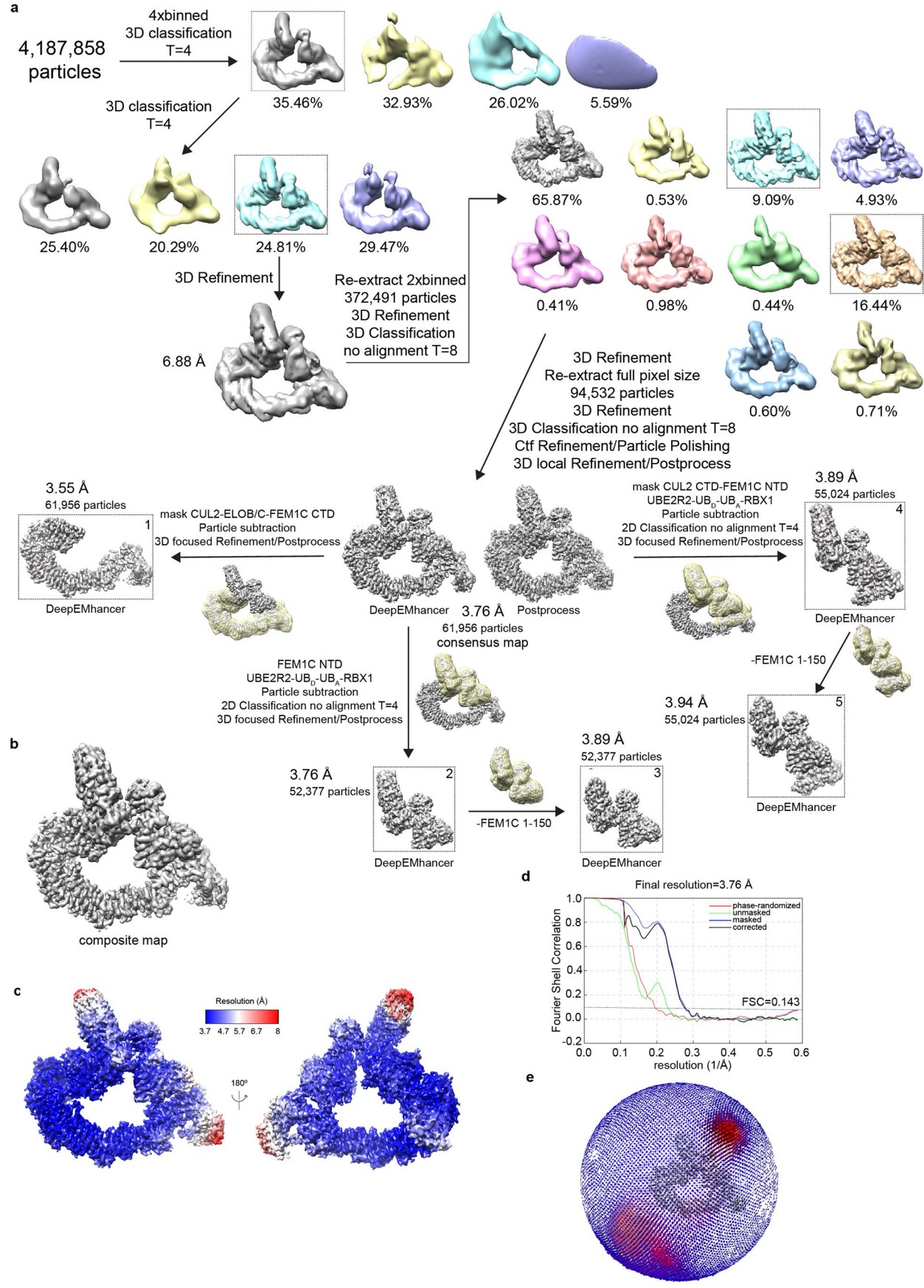

**Extended Data Fig. 2 | See next page for caption.**

**Extended Data Fig. 2 | Cryo-EM structure determination of the CRL poly-ubiquitin chain formation complex. a**, Flow chart describing the processing of the cryo-EM dataset (also see Table 1). The consensus map was refined to 3.76 Å. Additionally, five distinct focused maps were generated, with masking on the following subunits and the indicated residue ranges: (1) CUL2-Elongin B/C-FEM1C (residues 404-C)-RBX1 (residues 5-35); (2) FEM1C (residues N-404)-UBE2R2-UB$_D$-UB$_A$-Sil1 peptide-RBX1 (residues 35-104); (3) FEM1C (residues 150-404)-UBE2R2-UB$_D$-UB$_A$-Sil1-RBX1 (residues 35-104); (4) CUL2 (residues 429-556)-FEM1C (residues N-404)-UBE2R2-UB$_D$-UB$_A$-Sil1-RBX1 (residues 30-104); and (5) CUL2 (residues 429-556)-FEM1C (residues 150-404)-UBE2R2-UB$_D$-UB$_A$-Sil1-RBX1 (residues 30-104). N or C refer to N- or C-terminal residues. **b**, Final image of the composite cryo-EM map generated by merging all of the focused maps. **c**, Cryo-EM density of the consensus map as colored by the indicated local resolution. **d**, Graph plotting the Fourier shell correlation (FSC) versus inverse resolution (3.76 Å resolution at FSC = 0.143). **e**, Angular distribution of the consensus map. UB$_D$, donor ubiquitin; UB$_A$, acceptor ubiquitin.

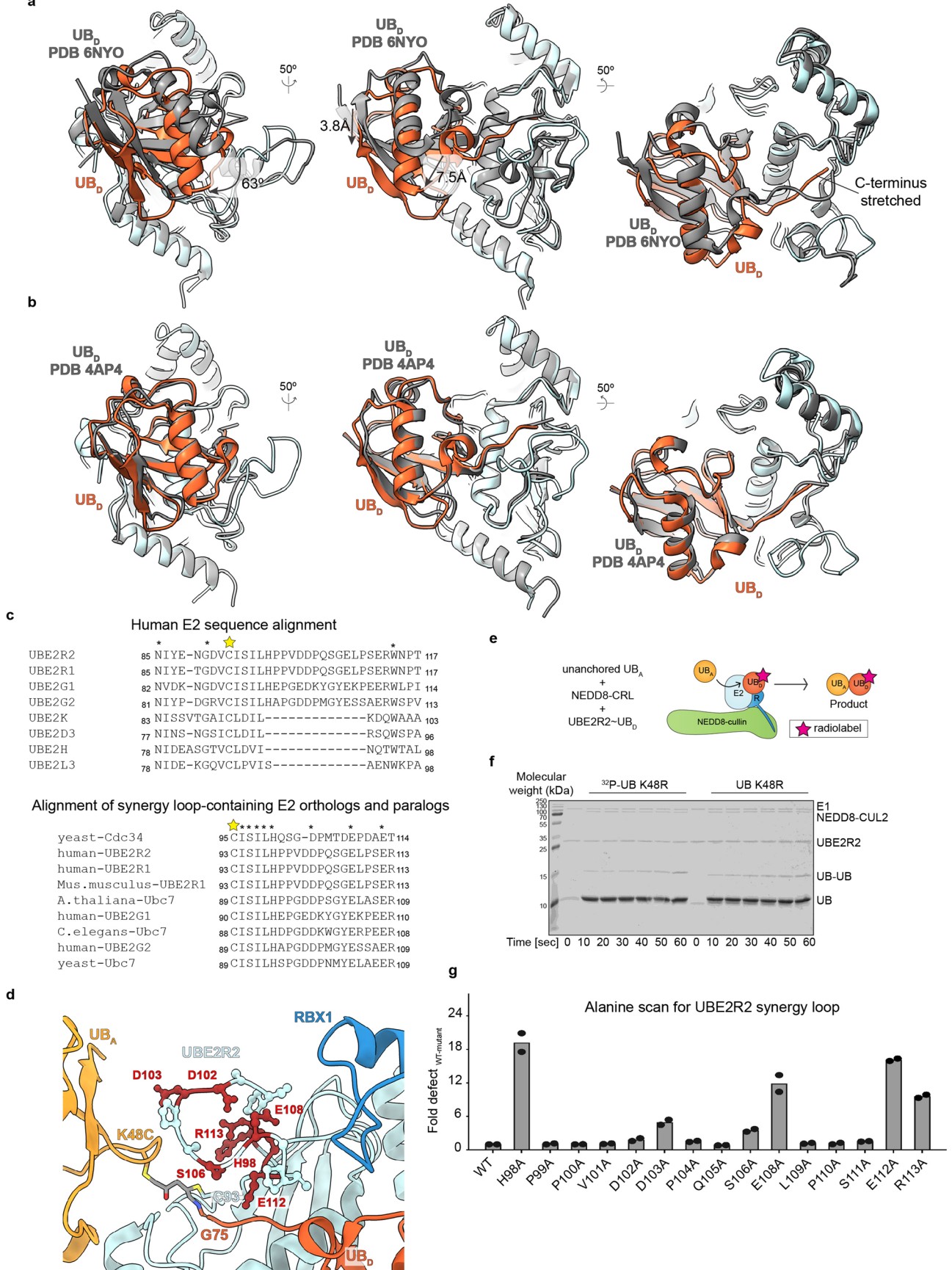

**Extended Data Fig. 3 | See next page for caption.**

**Extended Data Fig. 3 | The CRL poly-ubiquitin chain formation structure shows donor ubiquitin in an activated conformation. a**, Superposition of UBE2R2-donor ubiquitin ($UB_D$) from the neddylated CRL2$^{FEMIC-SiI1-UB}_A$-UBE2R2-$UB_D$ structure (cyan and dark orange, respectively) with the previously published x-ray structure of UBE2R2 - $UB_D$ bound to an allosteric inhibitor (gray; PDB code 6NYO). The structural alignment was performed on the E2 UBC domains. **b**, Same as in **a**, except with UBE2D1 - $UB_D$ bound to the RNF4 E3 RING domain (not shown). Also see Supplementary Video 2. **c**, Multiple sequence alignment of various human E2s (top). The star indicates the position of the catalytic Cys. Fully conserved residues have been labeled with an asterisk. A multiple sequence alignment of Cdc34/UBE2R and Ubc7/UBE2G-family E2s highlighting conservation of various positions in the synergy loop has also been provided (bottom). **d**, Ribbon diagram highlighting the positions of synergy loop residues (shown as ball-and-sticks). Residues colored red were shown upon their mutation to be defective in poly-ubiquitin chain formation. **e**, Schematic of the UBE2R2 unanchored, di-ubiquitin chain synthesis assay in the presence of neddylated cullin-RBX1 (R) without substrate receptors. Donor ubiquitin ($UB_D$) and acceptor ubiquitin ($UB_A$) are covalently joined ($UB_D$-$UB_A$) to form product. **f**, Coomassie-stained SDS-PAGE gel comparing di-ubiquitin product formation for $^{32}$P-labeled K48R ubiquitin (that contained an N-terminal tag that enabled phosphorylation by protein kinase A) and unlabeled K48R ubiquitin without a tag. The gel is representative of duplicate technical replicates. UB, ubiquitin. **g**, Bar graph showing fold defects in di-ubiquitin chain synthesis for synergy loop mutants in comparison with wild-type. The value of each bar represents the estimated value of 'fold-defect' based on n = 2 technical replicates.

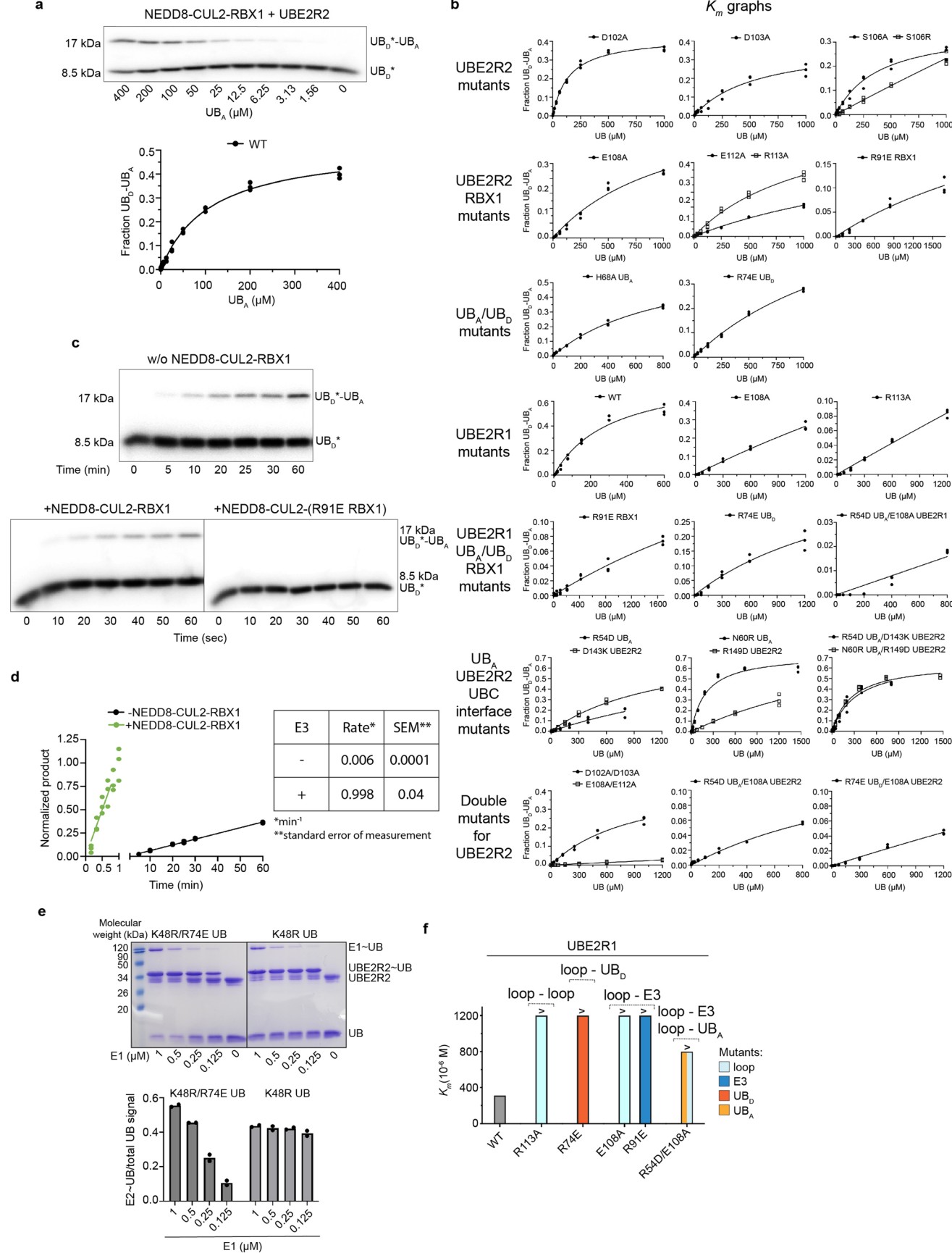

**Extended Data Fig. 4 | See next page for caption.**

**Extended Data Fig. 4 | Unanchored di-ubiquitin chain synthesis enables estimation of the $K_m$ and the apparent affinity of acceptor ubiquitin for UBE2R-family E2s. a**, Autoradiogram showing $UB_D$-$UB_A$ formation for reactions containing UBE2R2, neddylated CUL2-RBX1, and increasing $UB_A$ concentrations. The gel migrations of radiolabeled $UB_D$ (indicated by *) and $UB_A$-$UB_D$ product are shown (top). The graph shows the fraction of $UB_A$-$UB_D$ product formed as a function of the unanchored $UB_A$ concentration with wild-type (WT) components (bottom). The data were fit to the Michaelis-Menten model (GraphPad Prism software v9) to estimate the $K_m$ of $UB_A$ for the UBE2R2-mediated poly-ubiquitylation complex. The autoradiogram is representative of triplicate technical replicates. $UB_D$, donor ubiquitin; $UB_A$, acceptor ubiquitin. **b**, Graphs showing the fraction of di-ubiquitin product formation ($UB_D$-$UB_A$) as a function of the unanchored acceptor ubiquitin ($UB_A$) concentration with WT or mutant protein components (the assay setup is shown in Extended Data Fig. 3e). The data were fit to the Michaelis-Menten model using non-linear regression (GraphPad Prism software v9). Datapoints from triplicate technical replicates are shown. **c**, Autoradiograms showing time courses for the formation of free

$UB_A$-$UB_D$ product with WT UBE2R2. Reactions were performed in the absence (w/o) or presence of WT neddylated CUL2-RBX1 or a mutant harboring a R91E RBX1 subunit. All autoradiograms are representative of triplicate technical replicates. **d**, Graph showing product formation normalized to the ratio of the $UB_D$ and UBE2R2 concentrations with respect to time. Rates were derived by linear regression in Prism. Estimates are based on n = 3 technical replicates. **e**, Coomassie-stained SDS-PAGE gel comparing the loading of WT UBE2R2 with K48R or K48R/R74E donor ubiquitins and upon titration of E1 enzyme (top). The graph shows the fraction of the total ubiquitin thioesterified to UBE2R2 (bottom; based on n = 2 technical replicates). The results determined the E1 concentration in subsequent experiments with K48R/R74E donor ubiquitin (see methods). The gel is representative of duplicate technical replicates. UB, ubiquitin. **f**, Bar graphs comparing the $K_m$ values of unanchored $UB_A$ for UBE2R1 in the presence of neddylated CUL2-RBX1 and the indicated mutants. Bars showing a '>' reflect reactions where saturation of UBE2R2 with $UB_A$ was not feasible (the top concentration in the dilution series is shown). The value of each bar represents the estimated value for $K_m$ based on n = 3 technical replicates.

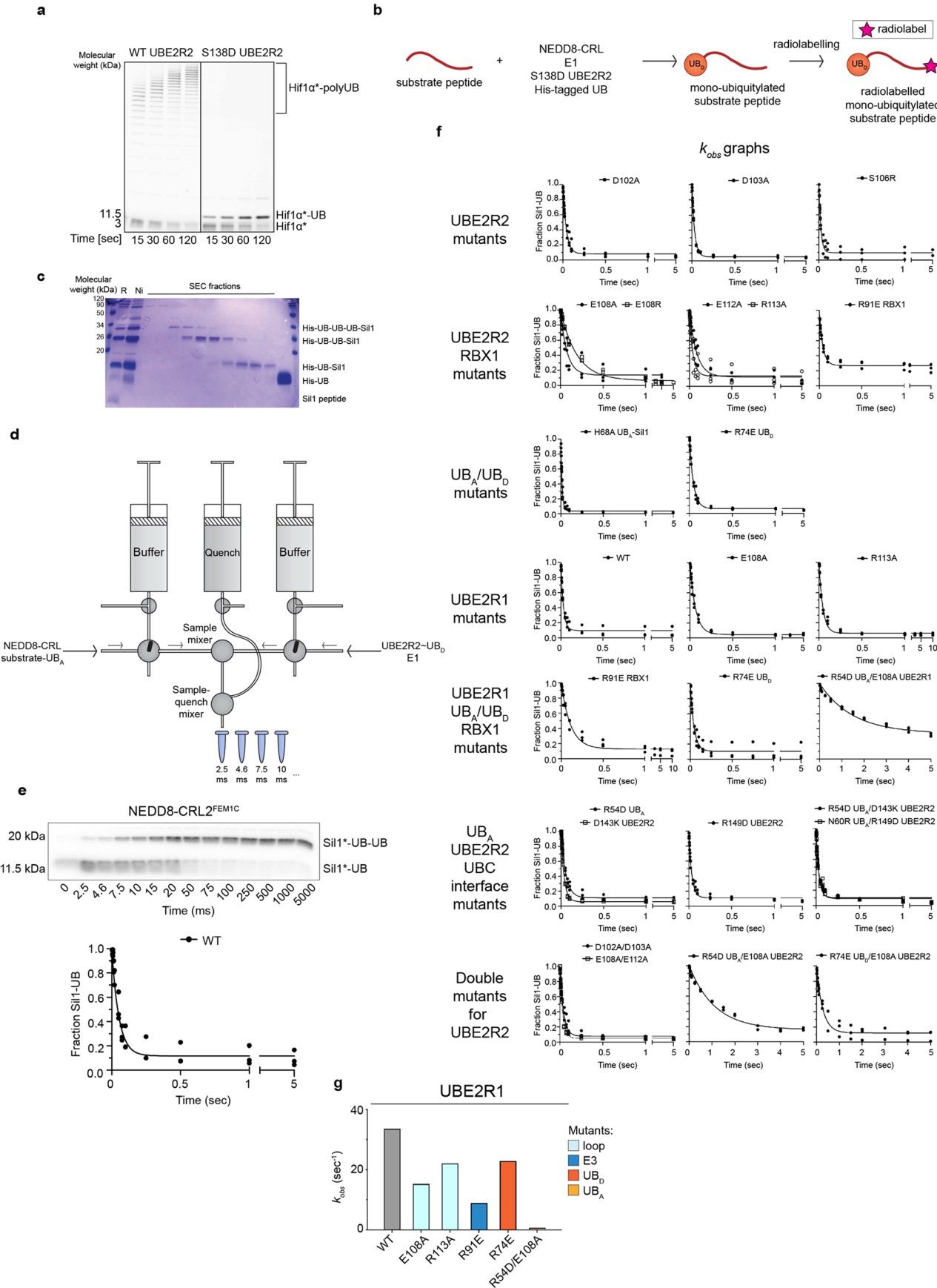

**Extended Data Fig. 5 | See next page for caption.**

**Extended Data Fig. 5 | Ubiquitin-primed CRL substrate ubiquitylation enables estimation of the rate of poly-ubiquitin chain formation $k_{obs}$.**
**a**, Fluorescence-scanned gel showing ubiquitylation reactions comparing wild-type (WT) and S138D UBE2R2 activity with Hif1α peptide substrate (see methods). Notice that, while S138D UBE2R2 is highly defective at poly-ubiquitin chain formation, it can prime substrate, facilitating production of highly pure Sil1-ubiquitin substrate. The scan is representative of duplicate technical replicates. UB, ubiquitin. **b**, Schematic of the production strategy for substrate-ubiquitin synthesis. **c**, Coomassie-stained SDS-PAGE gel showing purification of Sil1-ubiquitin. Following the ubiquitylation reaction (R), Sil1-ubiquitin was first separated from unreacted peptide owing to ubiquitin's N-terminal Histidine tag (Ni) followed by gel filtration chromatography. The gel is representative of triplicate technical replicates. SEC, size exclusion chromatography. **d**, Illustration of the Rapid Quench Flow (RQF) device (Kintek) used to estimate the rates of poly-ubiquitin chain formation ($k_{obs}$) on ubiquitin-primed CRL substrates by UBE2R2. Timepoints are computer-controlled according to the

speed of a plate pushing against the syringe plungers, first mixing the reactants followed by introduction of the quench buffer. **e**, Autoradiogram showing a time course from quench flow, pre-steady state kinetic ubiquitylation reactions with neddylated CRL2$^{FEMIC}$. The gel migrations of radiolabeled, ubiquitin-primed Sil1 substrate (indicated by *) and ubiquitylated product are shown (top). The graph shows the fraction of remaining substrate as a function of time in the presence of WT proteins (bottom). The autoradiogram is representative of triplicate technical replicates. **f**, Graphs showing the fraction of ubiquitin-primed Sil1 substrate remaining as a function of time with WT or mutant proteins. The data were fit to a single exponential decay model (GraphPad Prism software v9). Datapoints from triplicate technical replicates are shown. **g**, Bar graph comparing the ubiquitin transfer rates ($k_{obs}$) for the indicated proteins in reactions that contained neddylated CRL2$^{FEMIC}$, Sil1-ubiquitin substrate, and UBE2R1. The value of each bar represents the estimated value of $k_{obs}$ based on n = 3 technical replicates. UB$_D$, donor ubiquitin; UB$_A$, acceptor ubiquitin.

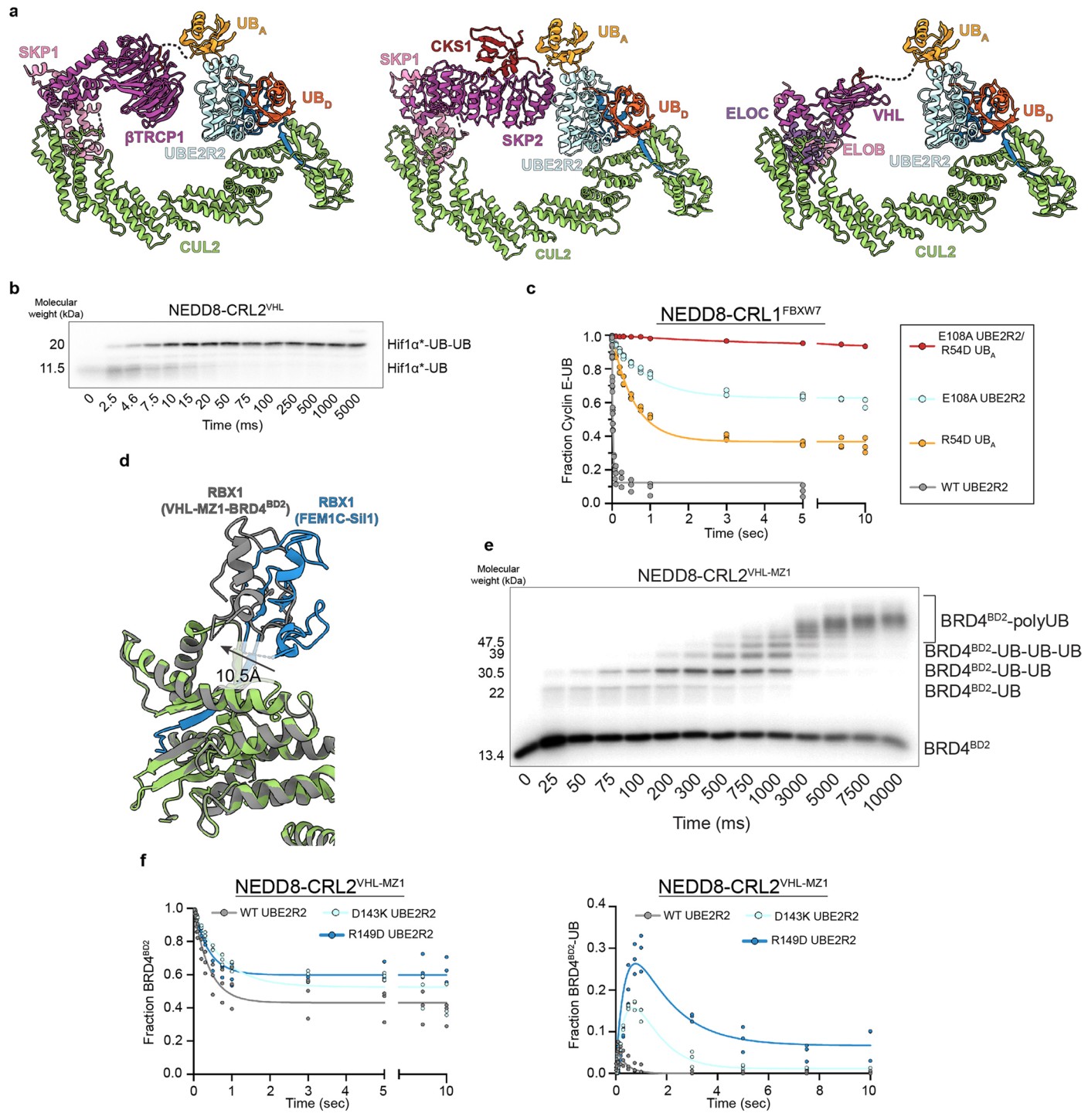

**Extended Data Fig. 6 | Generalized mechanism for CRL-dependent substrate poly-ubiquitylation. a,** Modelling of the RBX1-UBE2R2 - $UB_D$-$UB_A$ portion of the chain formation structure (PDB code: 8PQL, this work) onto three other CRL substrate receptor complexes: SKP1-βTRCP1 (left; PDB code 6TTU), SKP1-SKP2 (middle; PDB code 7B5L), and Elongin B/C-VHL (right; PDB code 1LM8). **b,** Autoradiogram from quench flow, pre-steady state kinetic ubiquitylation reactions with neddylated CRL2VHL and ubiquitin-primed Hif1α peptide substrate that had been radiolabeled (indicated by *). The graph showing substrate conversion to product and the fit of the data to the model can be found in Extended Data Fig. 7a. The autoradiogram is representative of triplicate technical replicates. UB, ubiquitin. **c,** Graph corresponding to ubiquitylation reactions containing neddylated CRL1FBXW7 with wild-type (WT) or mutant ubiquitin-primed cyclin E peptide substrates and WT or mutant UBE2R2. The data were fit to closed form solutions using Mathematica[19] (v.13.1). Datapoints from

triplicate technical replicates are shown. **d,** Superposition of the CUL2 C-terminal domains from the CRL2VHL-MZ1-BRD4-UBA-UBE2R2-UBD (gray) and CRL2FEM1C-Sil1-UBA-UBE2R2-UBD structures (green and blue), showing translation of the RBX1 RING domains that enables repositioning of UBE2R2-ubiquitin relative to substrate. **e,** Autoradiogram for BRD4BD2 ubiquitylation reactions in the presence of neddylated CRL2VHL, the PROTAC MZ1, UBE2R2 and WT ubiquitin. Notice the sequential appearance of ubiquitin-primed BRD4 (BRD4BD2-UB) and poly-ubiquitylated BRD4 (BRD4BD2-UB-UB) which enabled estimation of the rate of ubiquitin transfer to primed neo-substrate (Fig. 4e and Extended Data Table 1; see methods). The autoradiogram is representative of triplicate technical replicates. **f,** Graphs showing depletion of BRD4BD2 (left) or the appearance of primed neo-substrate (right) in the presence of the indicated UBE2R2 proteins and the fit of the data to the model. $UB_A$, acceptor ubiquitin; $UB_D$, donor ubiquitin; BRD4BD2, BRD4 (residues 346-460); ELOB, Elongin B; ELOC, Elongin C.

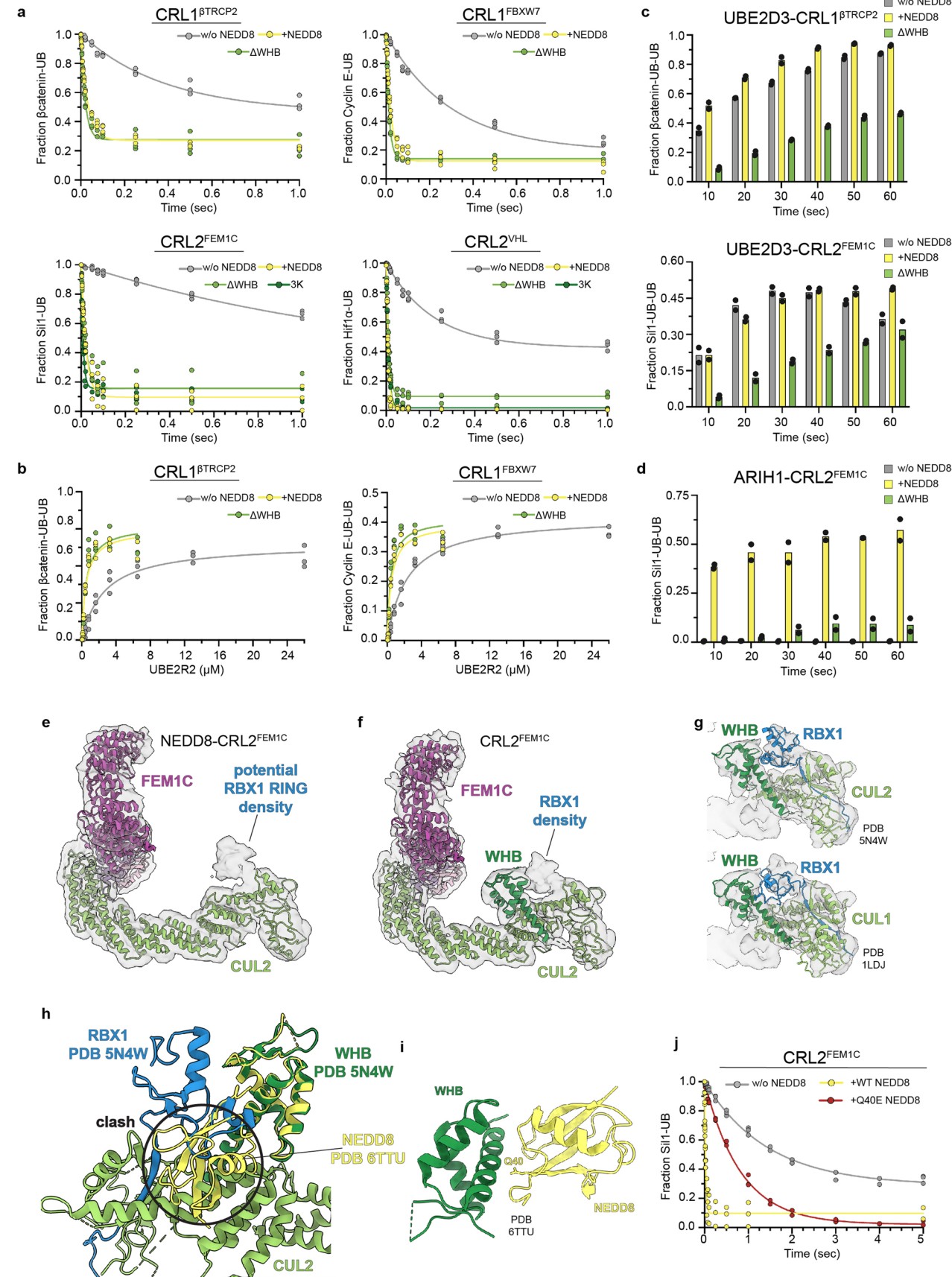

**Extended Data Fig. 7 | See next page for caption.**

**Extended Data Fig. 7 | NEDD8 activates CRLs for millisecond poly-ubiquitin chain formation by releasing RBX1's RING domain from cullin. a**, Graphs showing the fraction of ubiquitin-primed peptide substrates remaining with unneddylated CRL (w/o NEDD8, gray), neddylated CRL ( + NEDD8, yellow), unneddylated CRL with deletion of the cullin WHB domain (ΔWHB, light green), and CRL2s containing D660K E664K D675K CUL2 (3 K, dark green). CRL1$^{\beta TRCP2}$ (top-left), CRL1$^{FBXW7}$ (top-right), CRL2$^{FEM1C}$ (bottom-left) and CRL2$^{VHL}$ (bottom-right) E3s were examined. UB, ubiquitin. **b**, Graphs showing the fraction of ubiquitin-primed peptide substrates converted to ubiquitylated product as a function of UBE2R2 levels in the presence of unneddylated CRL (w/o NEDD8, gray), neddylated CRL ( + NEDD8, yellow), and unneddylated CRL with deletion of the CUL1 WHB domain (ΔWHB, light green). CRL1$^{\beta TRCP2}$ (left) and CRL1$^{FBXW7}$ (right) E3s were examined. **c**, Bar graphs showing the fraction of ubiquitin-primed peptide substrates converted to product in the presence of the E2 UBE2D3 and the indicated CRL complexes. CRL1$^{\beta TRCP2}$ (top) and CRL2$^{FEM1C}$ (bottom) E3s were examined. **d**, Same as **c**, except with the E3s ARIH1 and CRL2$^{FEM1C}$. **e**, Cryo-EM map of the neddylated CRL2$^{FEM1C}$ complex in the absence of UBE2R2 selected amongst

multiple reconstructions. Ribbon diagrams of models corresponding to Elongin B/C-FEM1C (this study) and CUL2 (PDB code 5N4W) were fit into the density. Note the presence of additional electron density potentially corresponding to the RBX1 RING domain. **f**, Same as in **e**, except showing a cryo-EM map of the unneddylated CRL2$^{FEM1C}$ complex. **g**, Same as in **f** except with ribbon diagrams of CUL2-RBX1 (top; PDB code 5N4W) or CUL1-RBX1 (bottom; PDB code 1LDJ) fit into the density. **h**, Superposition of CUL2-RBX1 (green and blue, respectively; PDB code 5N4W) onto the structure of a neddylated CRL1 bound to UBE2D (PDB code 6TTU, yellow). Severe steric clashing may be observed between NEDD8 and CUL2. The structural alignment was performed on the WHB domains. **i**, Ribbon diagram showing CUL1's WHB domain (green) in complex with conjugated NEDD8 (yellow; PDB code 6TTU). The position of NEDD8's Gln40 residue is shown. **j**, Graph corresponding to ubiquitylation reactions with ubiquitin-primed Sil1 peptide substrate and either unneddylated CRL2$^{FEM1C}$ (w/o NEDD8, gray), neddylated CRL2$^{FEM1C}$ (yellow), or neddylated CRL2$^{FEM1C}$ harboring Q40E NEDD8 (red). Graphs are representative of triplicate (**a,b** and **j**) or duplicate (**c** and **d**) technical replicates. w/o, without.

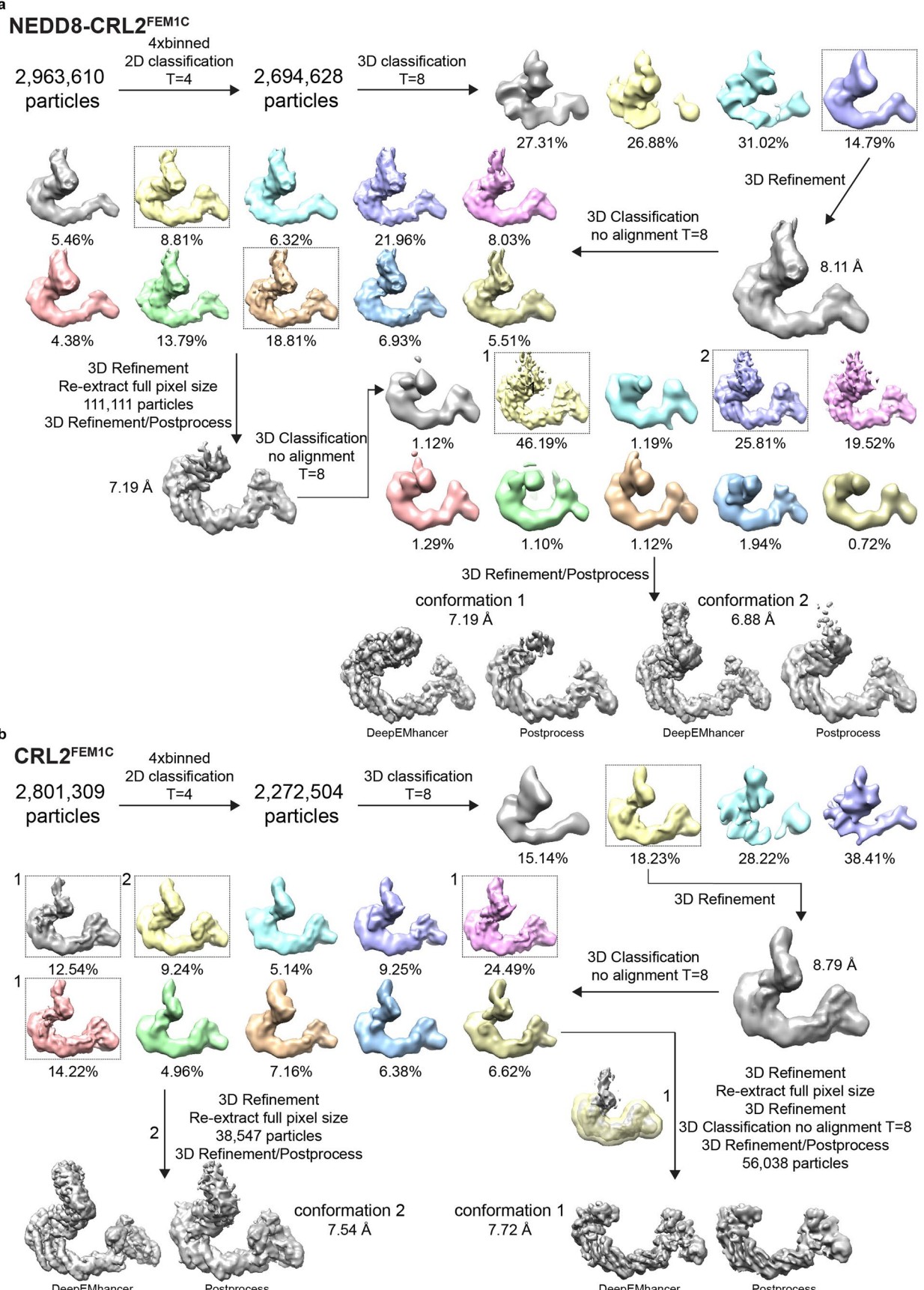

**Extended Data Fig. 8 | Cryo-EM structure determination of unneddylated or neddylated CRL2^FEM1C complexes. a**, Flow chart showing the processing of the cryo-EM dataset for the neddylated CRL2^FEM1C complex (also see Table 1). Two major conformations that were apparent during classification are depicted. **b**, same as in **a**, except for the unneddylated CRL2^FEM1C complex.

**Extended Data Table 1 | Estimates of $K_m$ and $k_{obs}$ for CRL1/2-mediated poly-ubiquitin chain formation**

| CRL | Substrate[a] | UBE2R2 | $K_m{}^b$ ($10^{-9}$ M) | $k_{obs}{}^{S1-S2}$ (sec$^{-1}$) | Fold change ($k_{obs}{}^{S1-S2}$) |
|---|---|---|---|---|---|
| NEDD8-CRL2$^{FEM1C}$ | Sil1-Ub$^{WT}$ | WT | - | 45.55 ± 2.88 | - |
| NEDD8-CRL2$^{VHL}$ | Hif1α-Ub$^{WT}$ | WT | - | 99.82 ± 5.49 | - |
| NEDD8-CRL1$^{FBXW7}$ | Cyclin E-Ub$^{WT}$ | WT | 480 ± 86 | 43.45 ± 1.99 | - |
| NEDD8-CRL1$^{βTrCP2}$ | βcatenin-Ub$^{WT}$ | WT | 504 ± 94 | 32.92 ± 1.80 | - |
| **Interface mutants** | | | | | |
| NEDD8-CRL1$^{FBXW7}$ | Cyclin E-Ub$^{WT}$ | E108A | - | 0.53 ± 0.04 | 82.0 |
| NEDD8-CRL1$^{FBXW7}$ | Cyclin E-Ub$^{R54D}$ | WT | - | 1.38 ± 0.08 | 31.5 |
| NEDD8-CRL1$^{FBXW7}$ | Cyclin E-Ub$^{R54D}$ | E108A | - | 0.02 ± 0.003 | 2173 |
| **CRL2$^{VHL-MZ1}$-BRD4$^{BD2}$ complex** | | | | | |
| NEDD8-CRL2$^{VHL}$ | BRD4$^{BD2}$-MZ1[c] | WT | | 24.97 ± 2.05 | - |
| NEDD8-CRL2$^{VHL}$ | BRD4$^{BD2}$-MZ1[c] | D143K | - | 1.65 ± 0.22 | 15.1 |
| NEDD8-CRL2$^{VHL}$ | BRD4$^{BD2}$-MZ1[c] | R149D | - | 0.57 ± 0.23 | 43.8 |
| **Un-neddylated CRL complexes** | | | | | |
| CRL2$^{FEM1C}$ | Sil1-Ub$^{WT}$ | WT | - | 0.70 ± 0.03 | 65.1 |
| CRL2$^{VHL}$ | Hif1α-Ub$^{WT}$ | WT | - | 4.68 ± 0.24 | 21.3 |
| CRL1$^{FBXW7}$ | Cyclin E-Ub$^{WT}$ | WT | 2366 ± 261 | 3.32 ± 0.13 | 13.1 |
| CRL1$^{βTrCP2}$ | βcatenin-Ub$^{WT}$ | WT | 2617 ± 433 | 2.58 ± 0.26 | 12.8 |
| **ΔWHB CRL complexes** | | | | | |
| ΔWHB-CRL2$^{FEM1C}$ | Sil1-Ub$^{WT}$ | WT | - | 49.61 ± 3.88 | 0.9 |
| ΔWHB-CRL2$^{VHL}$ | Hif1α-Ub$^{WT}$ | WT | - | 86.21 ± 3.50 | 1.2 |
| ΔWHB-CRL1$^{FBXW7}$ | Cyclin E-Ub$^{WT}$ | WT | 410 ± 98 | 74.80 ± 2.05 | 0.6 |
| ΔWHB-CRL1$^{βTrCP2}$ | βcatenin-Ub$^{WT}$ | WT | 517 ± 123 | 48.97 ± 2.90 | 0.7 |
| **3K CRL complexes** | | | | | |
| 3K-CRL2$^{FEM1C}$ | Sil1-Ub$^{WT}$ | WT | - | 114.52 ± 8.06 | 0.4 |
| 3K-CRL2$^{VHL}$ | Hif1α-Ub$^{WT}$ | WT | - | 99.34 ± 2.65 | 1.0 |
| **Bacterial deamidation mimic** | | | | | |
| (Q40E NEDD8)-CRL2$^{FEM1C}$ | Sil1-Ub$^{WT}$ | WT | - | 1.21 ± 0.03 | 37.6 |

[a]All substrates are peptides except BRD4$^{BD2}$ which is a protein; [b]$K_m$ of UBE2R2 for the CRL; [c]MZ1 is a PROTAC targeting BRD4 neo-substrate through the von Hippel-Lindau (VHL) CRL substrate receptor; BRD4$^{BD2}$, BRD4 (346-460); 3K, D660K E664K D675K CUL2-RBX1. The standard error of measurements are shown for all estimates.

Gary Kleiger

# Reporting Summary

## Statistics

For all statistical analyses, confirm that the following items are present in the figure legend, table legend, main text, or Methods section.

| n/a | Confirmed | |
|---|---|---|
| ☐ | ☒ | The exact sample size (*n*) for each experimental group/condition, given as a discrete number and unit of measurement |
| ☐ | ☒ | A statement on whether measurements were taken from distinct samples or whether the same sample was measured repeatedly |
| ☒ | ☐ | The statistical test(s) used AND whether they are one- or two-sided *Only common tests should be described solely by name; describe more complex techniques in the Methods section.* |
| ☒ | ☐ | A description of all covariates tested |
| ☒ | ☐ | A description of any assumptions or corrections, such as tests of normality and adjustment for multiple comparisons |
| ☐ | ☒ | A full description of the statistical parameters including central tendency (e.g. means) or other basic estimates (e.g. regression coefficient) AND variation (e.g. standard deviation) or associated estimates of uncertainty (e.g. confidence intervals) |
| ☒ | ☐ | For null hypothesis testing, the test statistic (e.g. *F*, *t*, *r*) with confidence intervals, effect sizes, degrees of freedom and *P* value noted *Give P values as exact values whenever suitable.* |
| ☒ | ☐ | For Bayesian analysis, information on the choice of priors and Markov chain Monte Carlo settings |
| ☒ | ☐ | For hierarchical and complex designs, identification of the appropriate level for tests and full reporting of outcomes |
| ☒ | ☐ | Estimates of effect sizes (e.g. Cohen's *d*, Pearson's *r*), indicating how they were calculated |

*Our web collection on statistics for biologists contains articles on many of the points above.*

## Software and code

Policy information about availability of computer code

| Data collection | Cryo-EM: SerialEM v3.8.0-b5. Gel imaging: Amersham Imager 600, Amersham Typhoon |
|---|---|
| Data analysis | Cryo-EM: RELION v3.1.1, MotionCorr2 v. 1.1.0, CryoSparc v4.2.0, CTFFIND v4.1, and Gautomatch v0.56. Structure Analysis and Visualization: Chimera v1.15 and ChimeraX v1.4. Model Building: COOT v0.9.6, Phenix.refine v1.19.2-4158, DeepEMhancer (https://github.com/rsanchezgarc/deepEMhancer), and AlphaFold2. Crosslinking mass spectrometry: Proteome Discoverer v2.5.0.400. Biochemistry: Prism v9 (Graphpad), Mathematica v13.1 (Wolfram), ImageQuant v8.2.0.0 (Cytiva). |

For manuscripts utilizing custom algorithms or software that are central to the research but not yet described in published literature, software must be made available to editors and reviewers. We strongly encourage code deposition in a community repository (e.g. GitHub). See the Nature Portfolio guidelines for submitting code & software for further information.

## Data

Policy information about availability of data

All manuscripts must include a data availability statement. This statement should provide the following information, where applicable:

- Accession codes, unique identifiers, or web links for publicly available datasets
- A description of any restrictions on data availability
- For clinical datasets or third party data, please ensure that the statement adheres to our policy

The atomic coordinates and electron microscopy maps have been deposited in the PDB with accession code 8PQL and in the Electron Microscopy Data Bank with codes EMD-17803 (consensus map) and EMD-17822 (composite map) for the neddylated CRL2FEM1C poly-ubiquitin chain formation complex, EMD-17802 for the neddylated CRL1FBXW7 poly-ubiquitin chain formation complex, EMD-18767 for the neddylated CRL2VHL-MZ1-BRD4 poly-ubiquitin chain formation complex, EMD-17798 and EMD-17799 for the CRL2FEM1C complex, and EMD-17800 and EMD-17801 for the neddylated CRL2FEM1C complex. Publicly available PDB entries are 1LDJ, 1LM8, 4AP4, 5AIT, 5N4W, 6LBN, 6NYO, 6TTU, 7B5L, 7MEY, and 7OJX. Source data are provided with this paper.

## Research involving human participants, their data, or biological material

Policy information about studies with human participants or human data. See also policy information about sex, gender (identity/presentation), and sexual orientation and race, ethnicity and racism.

| | |
|---|---|
| Reporting on sex and gender | No research involving human participants has been performed |
| Reporting on race, ethnicity, or other socially relevant groupings | No research involving human participants has been performed |
| Population characteristics | No research involving human participants has been performed |
| Recruitment | No research involving human participants has been performed |
| Ethics oversight | No research involving human participants has been performed |

Note that full information on the approval of the study protocol must also be provided in the manuscript.

# Field-specific reporting

Please select the one below that is the best fit for your research. If you are not sure, read the appropriate sections before making your selection.

☒ Life sciences          ☐ Behavioural & social sciences          ☐ Ecological, evolutionary & environmental sciences

For a reference copy of the document with all sections, see [nature.com/documents/nr-reporting-summary-flat.pdf](http://nature.com/documents/nr-reporting-summary-flat.pdf)

# Life sciences study design

All studies must disclose on these points even when the disclosure is negative.

| | |
|---|---|
| Sample size | N≥2. Sample sizes were chosen such that SEMs are typically 10% of the estimated values (and no more than 25%). |
| Data exclusions | no exclusions were made |
| Replication | N≥2 All experiments were performed independently. All attempts at replication were successful. |
| Randomization | Samples are not grouped as all comparisons are pairwise. |
| Blinding | Samples are not grouped such that blinding is not applicable to the study. |

# Reporting for specific materials, systems and methods

We require information from authors about some types of materials, experimental systems and methods used in many studies. Here, indicate whether each material, system or method listed is relevant to your study. If you are not sure if a list item applies to your research, read the appropriate section before selecting a response.

### Materials & experimental systems

| n/a | Involved in the study |
|---|---|
| ☒ | ☐ Antibodies |
| ☐ | ☒ Eukaryotic cell lines |
| ☒ | ☐ Palaeontology and archaeology |
| ☒ | ☐ Animals and other organisms |
| ☒ | ☐ Clinical data |
| ☒ | ☐ Dual use research of concern |
| ☒ | ☐ Plants |

### Methods

| n/a | Involved in the study |
|---|---|
| ☒ | ☐ ChIP-seq |
| ☒ | ☐ Flow cytometry |
| ☒ | ☐ MRI-based neuroimaging |

# Eukaryotic cell lines

Policy information about cell lines and Sex and Gender in Research

| | |
|---|---|
| Cell line source(s) | High five cell (BTI-TN-5B1-4) were obtained from ThermoFisher Scentific (catalogue number:B85502). Gibco Sf9 cells were obtained ThermoFisher Scentific (catalogue number:11496016). |
| Authentication | Cell lines were not authenticated. |
| Mycoplasma contamination | Cell lines were tested regularly for mycoplasma with no contamination detected. |
| Commonly misidentified lines (See ICLAC register) | No commonly misidentified cell lines were used in this study. |

