## [Peer Review File · Nature Structural & Molecular Biology]

Peer Review Information

Manuscript Title: Mechanism of millisecond Lys48-linked poly-ubiquitin chain formation by cullin-RING ligases

Corresponding author name(s): Brenda A. Schulman and Gary Kleiger

Reviewer Comments & Decisions:

Decision Letter, initial version:

Message: 29th Sep 2023

Dear Professor Kleiger,

Thank you again for submitting your manuscript "Mechanism of millisecond poly-ubiquitin chain formation by cullin-RING ligases". I apologise for the delay in responding, which resulted from the difficulty in obtaining suitable referee reports. Nevertheless, we now have comments (below) from the 3 reviewers who evaluated your paper. In light of these reports, we remain interested in your study and would like to see your response to the comments of the referees, in the form of a revised manuscript.

You will see that though reviewers #1 and #3 appreciate the novelty and importance of the structural and mechanistic findings, reviewer #2 voices concerns about whether these findings constitute enough of a conceptual advance to merit publication of the study in NSMB. We editorially think that experimentally performing the requested additional mutational analyses on UBE2R1 by reviewer #2, providing important textual clarifications, more concise explanation of certain findings, discussing the mechanistic questions posed, in accordance to the guidance of all experts, would significantly strengthen the manuscript. Additionally, we invite you to more clearly highlight the mechanistic novelty and potential wider implications of the study in relevant sections.

Please be sure to address/respond to all concerns of the referees in full in a point-by-point response and highlight all changes in the revised manuscript text file. If you have comments that are intended for editors only, please include those in a separate cover letter.

We expect to see your revised manuscript within 3-4 months. If you cannot send it within

this time, please contact us to discuss an extension; we would still consider your revision, provided that no similar work has been accepted for publication at NSMB or published elsewhere.

Reporting Summary:

When submitting the revised version of your manuscript, please pay close attention to our [href="https://www.nature.com/nature-portfolio/editorial-policies/image-integrity">Digital Image Integrity Guidelines. and to the following points below:](https://www.nature.com/nature-portfolio/editorial-policies/image-integrity)

Data availability: this journal strongly supports public availability of data. All data used in accepted papers should be available via a public data repository, or alternatively, as Supplementary Information. If data can only be shared on request, please explain why in

your Data Availability Statement, and also in the correspondence with your editor. Please note that for some data types, deposition in a public repository is mandatory - more information on our data deposition policies and available repositories can be found below: <https://www.nature.com/nature-research/editorial-policies/reporting-standards#availability-of-data>

[redacted]

Sincerely,

Dimitris Typas
Associate Editor
Nature Structural & Molecular Biology
ORCID: 0000-0002-8737-1319

Referee expertise:

Referee #1: CRL ubiquitin ligases-structural biology, UPS

Referee #2: CRL ubiquitin ligases-biochemistry and functional

Referee #3: E3 CRL ubiquitin ligases-structural biology

Reviewers' Comments:

Reviewer #1:

Remarks to the Author:

Cullin RING Ligases (CRLs) function with UBE2R family E2s to extend K48-polyUb chains on the millisecond time-scale, but the mechanism remains elusive. In this manuscript, the Kleiger and Schulman groups present the Cryo-EM structure of a trapped NEDDylated CUL2-RBX1-ELONGIN-B/C-FEM1C complex bound to UBE2R2-ubiquitin linked to a K48-ubiquitin-Sil1 peptide substrate, elucidating how this is achieved. E2 UBE2R2 contains a unique acid loop near its active site and a C-terminal acid tail. The structure reveals that UBE2R2's acid loop functions as a synergy loop that simultaneously engages RBX1, the donor ubiquitin, and the acceptor ubiquitin, optimizing the catalytic arrangement of K48-ubiquitin chain synthesis to maximize the catalytic efficiency of ubiquitin transfer. Mutagenesis and rapid quench flow analyses confirm the importance of these interactions during millisecond polyubiquitination. The study also unveiled an additional mechanism of NEDD8 activation of CRLs. In contrast to recent works from the Schulman group, where NEDD8 engages with the initiating E2 or E3 to facilitate the transfer of the first ubiquitin to the substrate, NEDD8 does not contact UBE2R2. NEDD8 modification leads to the release of the RBX1 RING domain, enabling RBX1 to optimally engage with UBE2R2 and promote millisecond polyubiquitination. This is an impressive work. The data are solid and the findings are exciting.

I only have few comments:

1. Prior work from Kleiger group suggested that the acidic loop might have a role in acceptor ubiquitin lysine deprotonation. The K48-acceptor ubiquitin/UBE2R2 Cys/donor ubiquitin active site was not shown here presumably disordered in the EM map. Figure 2a showed that acidic loop is close to the active site, could the authors comment on whether the acidic loop plays a role in positioning K48-acceptor ubiquitin?
2. The structure movie showed that UBE2R2 also contacts FEM1C. This was not investigated or discussed in the manuscript. Is this interaction relevant or is it specific for FEM1C?
3. In the extended Figure 1a, the authors showed that UBE2R2 1-187 lacking the C-terminal acidic tail was unable to form the trapped product. Could the authors visualize this tail in the Cryo-EM map? Also it would be helpful to label the bands in Extended Figure 1a.

Reviewer #2:

Remarks to the Author:

In this manuscript Liwocha et al. have utilized an activity-based probe to stabilize a transition state intermediate of a Cullin-2 RING ligase complex in which the ubiquitin chain-elongating enzyme UBE2R2 is captured in the act of ubiquitinating an already ubiquitin-primed substrate and determine its cryo-EM structure. This structure, coupled with resulting mutagenesis experiments, reveals the importance of the previously identified 'acidic loop' (renamed by the authors as the 'synergy loop') in ubiquitin chain elongation by UBE2R2. They find that this loop interacts simultaneously with RBX1, and both the acceptor and donor ubiquitins. The authors use rapid quench flow kinetics to reveal the remarkable speed with which the CRL/UBE2R2 pairing can ubiquitylate primed substrates and provide evidence that cullin neddylation promotes polyubiquitination by triggering a CRL remodeling that changes the position of the cullin's WHB domain, thereby allowing UBE2R2 to interact with the catalytic RBX1 subunit and acceptor ubiquitin.

The manuscript is well written, clear, and concise. The structures are well explained and all relevant controls included in experiments. My main concern is whether the findings represent a sufficient advance in our knowledge and are of sufficient interest to those outside the CRL field.

Minor points

- The authors present the structure of UBE2R2 catalyzing polyubiquitin chain formation. All mutagenesis experiments are performed using UBE2R2. The authors then conclude that both UBE2R1 and UBE2R2 work in the same manner. Would it not make sense to test some of their mutations in UBE2R1? For example, taking Figure 2F, are similar effects observed with equivalent mutations in UBE2R1?
- The UBE2R2 tail does not seem to be visible in the structure. Does this mean that this region is particularly flexible? Perhaps the authors could comment on the implications of this region for catalysis?
- The authors use UBE2R2 (S138D) for the generation of ubiquitin-primed peptide substrates. No explanation is given as to why this mutant is used. Could the authors please clarify?
- The authors use ³²P-labelled ubiquitin but provide no explanation as to its origin. Could the authors include details of its synthesis. Presumably some additional phosphorylation site is incorporated into the ubiquitin sequence and then phosphorylated with a relevant kinase? And could the presence of a phosphate group influence overall ubiquitination?
- Page 23, Line 695 mentions "32P-labeled peptide substrate". Could the authors please clarify whether the phosphate residues on the peptide or the associated ubiquitin, as it is not clear from the methods section.
- The authors use a CUL2 (D660K, E664K, D675K) mutant to liberate the WHB domain in the absence of neddylation and a NEDD8 (Q40E) mutant to achieve a similar effect. A figure is provided (Extended Data Fig 7i) to describe where the Q40E mutation lies and why it would disrupt the NEDD8-WHB interaction, but no equivalent figure is provided to explain the CUL2(3K) mutation. Perhaps the reader would find this helpful?
- It is notable that the NEDD8 Q40E mutation is a mimic of bacterial deamidation of the Q40 residue. Is this worth mentioning?

Reviewer #3:

Remarks to the Author:

Liwocha et al report through elegant structural and biochemical measurements the mechanism of poly ubiquitin chain extension by the E2 UBE2R1/2 in the presence of

neddylated CRLs. While numerous detailed mechanistic insights have been obtained for substrate priming, how ubiquitin chain extension by the Lys48 generating E2 family UBE2R has remained a mystery.

The authors build upon extensive structural expertise and utilisation of a chemical probe to mimic UBE2R1 in action assembling a Lys48-linked diubiquitin. Incorporating this trapped intermediate enable the authors to determine the atomic structure of neddylated CUL2-RBX1 bound to substrate receptor and a ubiquitinated substrate peptide and UBE2R1~Ub at an overall resolution 3.8 Å. Sufficient detail exists to unambiguously model high resolution structures in the map and importantly, sufficient information is obtained to provide positional side chain information at the corresponding CRL/RING/E2/ubiquitin interfaces.

The authors then use single point mutations to tease apart the importance of the unique UBE2R1/2 acidic loop (renamed synergy loop) in chain extension. Through a novel stopped flow setup, the authors were then able to record chain extensions mediated by CRL in the millisecond timeframe.

Disrupting mutations, and importantly rescue mutations, between the acceptor ubiquitin and UBE2R1 nicely demonstrate the importance of the newly observed interface for the acceptor ubiquitin binding to UBE2R1.

To see if RING engagement to UBE2R formed a common mechanism, the authors determined a second structure this time containing neddylated CUL1-RBX1 SKP1-FBXW7 with a CyclinE phospho-substrate ubiquitinated peptide bound to UBE2R. While the map was not of sufficient resolution for a new model to be built, the authors could convincingly show a common architecture that would enable enhancement of UBE2R1 chain elongation activity.

Surprisingly, the authors observed that the WHB domain of CRL1 and NEDD8 were flexible and not observed in the structures. While other RBX1 partner enzymes were required for effective substrate ubiquitination, deletion of the WHB domain from several CRLs did not impede UBE2R2 K48 chain enhancement. Through identification of a small subset of classes from maps derived from unneddylated CRL and comparison of crystal structures revealed a striking clash with the RBX1 RING. Satisfyingly, point mutations that prevented interaction between WHB and CUL2 enabled chain extension in the absence of neddylation.

In summary, this reviewer found the structural interpretation and biochemistry to be of very high standard. Perhaps the second complex of neddylated CUL1-RBX1 SKP1-FBXW7 could have been collected using a Krios to enable model building. However, given the synergy between cross-complex analysis and point mutations this is not a requirement. This reviewer only has a couple of minor comments detailed below and believes the article should be published in NSMB.

Minor comments

The description comparing the altered donor ubiquitin conformation on page 5 is confusing and somewhat misleading. This reviewer finds it difficult to compare the conformation of ubiquitin in the inhibited UBE2R1 crystal structure, where the authors propose a dramatic rearrangement. Firstly, the figure doesn't give the impression of a dramatic difference in

orientation of the ubiquitin between the two structures. Perhaps an alternative view or simply representing distances and angles between the two ubiquitins may help here. Secondly, as the authors acknowledge, this is compared to a crystal structure where lattice constraints may position the ubiquitin in a different conformation. Given the presence of an inhibitor and/or crystal packing may alter the conformation of the ubiquitin the authors may wish to tone down the paragraph on page 5.

In figure 2 the authors make mutations in the synergy loop to reveal the contribution of RBX1 to enhance the activity of UBE2R1 chain elongation activity. On the whole, mutations within the synergy loop impair K_m for unanchored acceptor ubiquitination. However, the greater mutations were from an interface on ubiquitin (R74E). Could this be due to a general impairment of UBE2R1 loading with ubiquitin? In addition, regarding the UBE2R1-interface would an E108R mutation show a comparable defect compared to RBX1 R91E?

The end of line 61 should have a reference to the 69 F-box proteins.

Author Rebuttal to Initial comments

Reviewer #1:

Remarks to the Author:

Cullin RING Ligases (CRLs) function with UBE2R family E2s to extend K48-polyUb chains on the millisecond time-scale, but the mechanism remains elusive. In this manuscript, the Kleiger and Schulman groups present the Cryo-EM structure of a trapped NEDDylated CUL2-RBX1-ELONGIN-B/C-FEM1C complex bound to UBE2R2-ubiquitin linked to a K48-ubiquitin-Sil1 peptide substrate, elucidating how this is achieved. E2 UBE2R2 contains a unique acid loop near its active site and a C-terminal acid tail. The structure reveals that UBE2R2's acid loop functions as a synergy loop that simultaneously engages RBX1, the donor ubiquitin, and the acceptor ubiquitin, optimizing the catalytic arrangement of K48-ubiquitin chain synthesis to maximize the catalytic efficiency of ubiquitin transfer. Mutagenesis and rapid quench flow analyses confirm the importance of these interactions during millisecond polyubiquitination. The study also unveiled an additional mechanism of NEDD8 activation of CRLs. In contrast to recent works from the Schulman group, where NEDD8 engages with the initiating E2 or E3 to facilitate the transfer of the first ubiquitin to the substrate, NEDD8 does not contact UBE2R2. NEDD8 modification leads to the release of the RBX1 RING domain, enabling RBX1 to optimally engage with UBE2R2 and promote millisecond polyubiquitination. This is an impressive work. The data are solid and the findings are exciting.

We thank the reviewer for these positive comments.

1. Prior work from Kleiger group suggested that the acidic loop might have a role in acceptor ubiquitin lysine deprotonation. The K48-acceptor ubiquitin/UBE2R2 Cys/donor ubiquitin active site was not shown here presumably disordered in the EM map. Figure 2a showed that acidic loop is close to the active site, could the authors comment on whether the acidic loop plays a role in positioning K48-acceptor ubiquitin?

We apologize for the lack of clarity. The revised manuscript now clearly states that the acidic loop (Asp103) helps place Lys48 into the active site through its proximity with acceptor ubiquitin's His68 (lines 176-177).

A Lys48 cannot be visualized since this residue was replaced with Cys to generate the chemically-stable complex between UBE2R2 and donor and acceptor ubiquitins. Nevertheless, we provide a new figure (Extended Data Fig. 1b) highlighting the UBE2R2 active site showing electron density for the synergy loop, the donor ubiquitin's C-terminal tail and the acceptor ubiquitin. This includes clear density for synergy loop residue His98, found to be important for UBE2R2 activity in the study referenced by the reviewer (Ziamba et al., *JBC*, 2013), whose side-chain imidazole ring points towards the UBE2R2 active site Cys93.

2. The structure movie showed that UBE2R2 also contacts FEM1C. This was not investigated or discussed in the manuscript. Is this interaction relevant or is it specific for FEM1C?

One of our major findings is that the geometry of the catalytic core - comprising UBE2R2~donor ubiquitin, acceptor ubiquitin, and RBX1's RING domain - is the same irrespective of the protein substrate, the cullin subunit (at least for CUL1 and CUL2), or the substrate receptor. In our initial submission, this point was supported by the cryo-EM structures representing UBE2R2-mediated Lys48-linked chain extension for a ubiquitin-primed Cyclin E phosphopeptide of CRL1^{FBXW7} (containing neddylated CUL1-RBX1 and substrate receptor SKP1-FBXW7) as well as Sil1 peptide recruited to CRL2^{FEM1C}. This conclusion is further bolstered in the revision, with a cryo-EM reconstruction with another system: a 'neo-substrate', BRD4 recruited via the PROTAC MZ1 to the ELONGIN B/C-VHL substrate receptor and neddylated CUL2-RBX1. The results (lines 255-273, Fig. 4, and supplemental movie 1) further establish generality for the findings.

In addition, the UBE2R2 interface with FEM1C is specific, which presumably explains the superior resolution of the cryo-EM data for this complex. We tested this system based on our biochemical data showing that neddylated CRL2^{FEM1C} preferentially functions with UBE2R2, both during ubiquitin chain elongation as well as substrate priming. The latter findings are described in a related manuscript that is now under consideration and is provided to the reviewers. Data in the substrate priming-focused paper show that the UBE2R2-FEM1C interface is important during substrate priming but not for poly-ubiquitin chain formation.

3. In the extended Figure 1a, the authors showed that UBE2R2 1-187 lacking the C-terminal acidic tail was unable to form the trapped product. Could the authors visualize this tail in the Cryo-EM map?

We have obtained cryo-EM reconstructions for five complexes with UBE2R2, the three showing the structural basis for poly-ubiquitylation reported in this manuscript, and two others showing substrate priming in our related manuscript under consideration. Despite extensive efforts that include but are not limited to intensive focused refinements, we did not observe the C-terminal acidic tail in any maps showing poly-ubiquitin chain formation. Density was observed for the tail for one of the two complexes representing substrate priming, localized in the cullin canyon as expected. However, despite the high resolution of the rest of that structure, we were unable to unambiguously model the tail sequence into the density. Overall, the data are consistent with ours and others' previous studies indicating that these interactions are relatively heterogeneous and/or dynamic.

Also it would be helpful to label the bands in Extended Figure 1a.

The bands have all been labeled and we thank the reviewer for the suggestion.

Reviewer #2:

Remarks to the Author:

In this manuscript Liwocha et al. have utilized an activity-based probe to stabilize a transition state intermediate of a Cullin-2 RING ligase complex in which the ubiquitin chain-elongating enzyme UBE2R2 is captured in the act of ubiquitinating an already ubiquitin-primed substrate and determine its cryo-EM structure. This structure, coupled with resulting mutagenesis experiments, reveals the importance of the previously identified 'acidic loop' (renamed by the authors as the 'synergy loop') in ubiquitin chain elongation by UBE2R2. They find that this loop interacts simultaneously with RBX1, and both the acceptor and donor ubiquitins. The authors use rapid quench flow kinetics to reveal the remarkable speed with which the CRL/UBE2R2 pairing can ubiquitylate primed substrates and provide evidence that cullin neddylation promotes polyubiquitination by triggering a CRL remodeling that changes the position of the cullin's WHB domain, thereby allowing UBE2R2 to interact with the catalytic RBX1 subunit and acceptor ubiquitin.

The manuscript is well written, clear, and concise. The structures are well explained and all relevant controls included in experiments.

We thank the reviewer for these positive comments.

My main concern is whether the findings represent a sufficient advance in our knowledge and are of sufficient interest to those outside the CRL field.

The work here serves as a paradigm for all E3s that build poly-ubiquitin chains onto protein substrates and provides insight into how any enzyme may evolve to display extraordinary catalytic efficiencies. We believe our study will be of broad interest for the following reasons:

- The human genome encodes roughly 300 different CRLs. Plants and some other organisms express nearly a thousand such complexes. Our manuscript describes for the first time how this large number of E3 ligases function - by catalyzing Lys48-linked poly-ubiquitin chain formation.
- Almost all facets of cell biology and regulation depend on Lys48-linked poly-ubiquitin chain formation by CRLs. We note in the revised Introduction that Lys48-linked poly-ubiquitin chains are the most prevalent chain type in all organisms investigated so far, including humans. Thus, understanding how they are formed on substrates is fundamentally important.
- From an even broader perspective, our study reveals two unexpected types of allosteric modulation underlying E3 ligase activity. First, neddylation activates poly-ubiquitylation by overcoming autoinhibition of CUL1 and CUL2. As noted by Reviewer 1, this is in striking contrast to the mechanism by which neddylation activates ubiquitylation by UBE2D-family E2s or ARIH1. Second, the RING domain allosterically activates E2 binding of the acceptor ubiquitin. This is a novel concept which we expect will be of interest to the many areas of research that involve ubiquitylation by E2-E3 complexes.
- CRLs are the predominant class of E3 employed for the targeted protein degradation drug discovery platform. Lys48-linked, poly-ubiquitylation underlies CRL-dependent targeted protein degradation. Knowledge of the structural basis of how these enzymes function will enable drug discovery.

In response to the reviewer and to further increase the impact of our study, we obtained a cryo-EM reconstruction illuminating poly-ubiquitylation during targeted protein degradation, specifically UBE2R2-mediated Lys48-linked poly-ubiquitin chain formation for a 'neo-substrate', BRD4 recruited via the PROTAC MZ1 to neddylated CRL2^{VHL}. Importantly, the catalytic core showing chain formation is consistent with the mechanism we reported in the original manuscript. These data are now shown as Fig. 4d.

- A large fraction of all RING-type E3 ligases function by coaxing a linkage-specific E2 to place ubiquitin chains onto an E3-bound substrate, yet to date there are no structures of human origin showing such a mechanism. Our data will establish paradigms for many E3 ligases outside the cullin-RING family as well.

We have edited the Introduction, Results, and Discussion sections to better reflect these concepts.

Minor points

- The authors present the structure of UBE2R2 catalyzing polyubiquitin chain formation. All mutagenesis experiments are performed using UBE2R2. The authors then conclude that both UBE2R1 and UBE2R2 work in the same manner. Would it not make sense to test some of their mutations in UBE2R1? For example, taking Figure 2F, are similar effects observed with equivalent mutations in UBE2R1?

We performed the requested experiments. All the mutations led to similar defects in the K_m of unanchored ubiquitin for UBE2R1 and k_{obs} , the rate of ubiquitin transfer to substrate, that were observed with UBE2R2-based E2s (please see Table 1 in the revised manuscript).

- The UBE2R2 tail does not seem to be visible in the structure. Does this mean that this region is particularly flexible? Perhaps the authors could comment on the implications of this region for catalysis?

We have obtained cryo-EM reconstructions for five complexes with UBE2R2, the three showing the structural basis for poly-ubiquitylation reported in this manuscript, and two others showing substrate priming in our related manuscript under consideration that we provide to the Reviewers. Despite extensive efforts that include but are not limited to intensive focused refinements, we did not observe the C-terminal acidic tail in any maps showing poly-ubiquitin chain formation. Density was observed for the tail for one of the two complexes representing substrate priming, localized in the cullin canyon as expected. However, despite the high resolution of the rest of that structure, we were unable to unambiguously model the tail sequence into the density. Overall, the data are consistent with ours and others' previous studies indicating that these interactions are relatively heterogeneous and/or dynamic.

In response to these comments, and related suggestions from Reviewer #1, we have added a paragraph to the revised Discussion section regarding the UBE2R2 tail.

- The authors use UBE2R2 (S138D) for the generation of ubiquitin-primed peptide substrates. No explanation is given as to why this mutant is used. Could the authors please clarify?

We thank the reviewer for pointing this out. To generate a ubiquitin-primed peptide substrate, one would ideally choose an E2 (or ARIH1) that are very efficient at substrate priming but not poly-ubiquitin chain formation. We had noticed in the literature that, while S138A UBE2R1 was defective at priming, it was even more defective at chain formation (Gazdoiu et al, *Molecular and Cellular Biology*, 2007). Those results inspired our testing other variants, including S138D UBE2R2, which was motivated by the work on the E2 Ubc9 showing that residues at the structurally equivalent position can be either Ser or Asp (Yunus et al, *Nature Structural and Molecular Biology*, 2006). We discovered that S138D UBE2R2 is still defective at poly-ubiquitin chain formation, but quite active for substrate priming (Extended Data Fig. 5a). This result formed the basis for our purification strategy (Extended Data Fig. 5b), where S138D UBE2R2 was used in reactions containing neddylated CRL2^{FEM1C}, Sil1 peptide, and His-tagged ubiquitin. Unreacted Sil1 peptide was first removed by isolation of the ubiquitylated products with Nickel-agarose resin, followed by gel filtration and the collection of fractions that contained pure Sil1-ubiquitin (Extended Data Fig. 5c). Since we were purifying a variety of ubiquitin-primed substrates, including those that contained mutations in the acceptor ubiquitin, we found that S138D UBE2R2 wasn't the most efficient in all cases (here the E2 UBE2D3 was used instead). The protocols are now described in detail in the methods section and the results are shown in the revised Extended Data Fig. 5a-c. Also, we wish to emphasize that this mutant was used only to produce the ubiquitin-primed substrate. The UBE2R2 in the structure has the native S138.

- The authors use ³²P-labelled ubiquitin but provide no explanation as to its origin. Could the authors include details of its synthesis. Presumably some additional phosphorylation site is incorporated into the ubiquitin sequence and then phosphorylated with a relevant kinase? And could the presence of a phosphate group influence overall ubiquitination?

The reviewer is correct that we have incorporated a tag at the N-terminus of ubiquitin that is recognized by protein kinase A. Based on the structures presented here and our observation that the donor ubiquitin N-terminus points away from the structure (please see figure below), we rationalized that tagging would not significantly affect UBE2R activity. To address the Reviewer comment, we tested this in di-ubiquitin synthesis assays (see schematic shown in Extended Data Fig. 3e) comparing unlabeled and untagged ubiquitin with ³²P-labeled donor ubiquitin. Di-ubiquitin product levels were similar across the time courses (Extended Data Fig. 3f). We now include a detailed description of the construct and our labeling protocol in the revised methods section.

- Page 23, Line 695 mentions “32P-labeled peptide substrate”. Could the authors please clarify whether the phosphate residues on the peptide or the associated ubiquitin, as it is not clear from the methods section.

We apologize for this oversight. We now state in the methods section that all peptides were synthesized with a 'RRSY' tag that enabled peptide phosphorylation by protein kinase A on the Ser residue.

- The authors use a CUL2 (D660K, E664K, D675K) mutant to liberate the WHB domain in the absence of neddylation and a NEDD8 (Q40E) mutant to achieve a similar effect. A figure is provided (Extended Data Fig 7i) to describe where the Q40E mutation lies and why it would disrupt the NEDD8-WHB interaction, but no equivalent figure is provided to explain the CUL2(3K) mutation. Perhaps the reader would find this helpful?

We agree with the reviewer and now include a figure (Fig. 6b) showing the location of the relevant residues relative to the WHB and cullin C/R domains as well as the RBX1 RING.

- It is notable that the NEDD8 Q40E mutation is a mimic of bacterial deamidation of the Q40 residue. Is this worth mentioning?

We agree. We have added the following statement to the revised manuscript (please also see lines 328-330):

"Notably, NEDD8's Gln40 is the primary target of pathogenic bacterial effector proteins that catalyze its deamidation, impairing CRL activity and ubiquitylation of their substrates"

Reviewer #3:

Remarks to the Author:

Liwocha et al report through elegant structural and biochemical measurements the mechanism of poly ubiquitin chain extension by the E2 UBE2R1/2 in the presence of neddylated CRLs. While numerous detailed mechanistic insights have been obtained for substrate priming, how ubiquitin chain extension by the Lys48 generating E2 family UBE2R has remained a mystery.

The authors build upon extensive structural expertise and utilisation of a chemical probe to mimic UBE2R1 in action assembling a Lys48-linked diubiquitin. Incorporating this trapped intermediate enable the authors to determine the atomic structure of neddylated CUL2-RBX1 bound to substrate

receptor and a ubiquitinated substrate peptide and UBE2R1~Ub at an overall resolution 3.8 Å. Sufficient detail exists to unambiguously model high resolution structures in the map and importantly, sufficient information is obtained to provide positional side chain information at the corresponding CRL/RING/E2/ubiquitin interfaces.

The authors then use single point mutations to tease apart the importance of the unique UBE2R1/2 acidic loop (renamed synergy loop) in chain extension. Through a novel stopped flow setup, the authors were then able to record chain extensions mediated by CRL in the millisecond timeframe.

Disrupting mutations, and importantly rescue mutations, between the acceptor ubiquitin and UBE2R1 nicely demonstrate the importance of the newly observed interface for the acceptor ubiquitin binding to UBE2R1.

To see if RING engagement to UBE2R formed a common mechanism, the authors determined a second structure this time containing neddylated CUL1-RBX1 SKP1-FBXW7 with a CyclinE phospho-substrate ubiquitinated peptide bound to UBE2R. While the map was not of sufficient resolution for a new model to be built, the authors could convincingly show a common architecture that would enable enhancement of UBE2R1 chain elongation activity.

Surprisingly, the authors observed that the WHB domain of CRL1 and NEDD8 were flexible and not observed in the structures. While other RBX1 partner enzymes were required for effective substrate ubiquitination, deletion of the WHB domain from several CRLs did not impede UBE2R2 K48 chain enhancement. Through identification of a small subset of classes from maps derived from unneddylated CRL and comparison of crystal structures revealed a striking clash with the RBX1 RING. Satisfyingly, point mutations that prevented interaction between WHB and CUL2 enabled chain extension in the absence of neddylation.

In summary, this reviewer found the structural interpretation and biochemistry to be of very high standard. Perhaps the second complex of neddylated CUL1-RBX1 SKP1-FBXW7 could have been collected using a Krios to enable model building. However, given the synergy between cross-complex analysis and point mutations this is not a requirement. This reviewer only has a couple of minor comments detailed below and believes the article should be published in NSMB.

Thank you!

Minor comments

The description comparing the altered donor ubiquitin conformation on page 5 is confusing and somewhat misleading. This reviewer finds it difficult to compare the conformation of ubiquitin in the inhibited UBE2R1 crystal structure, where the authors propose a dramatic rearrangement. Firstly, the figure doesn't give the impression of a dramatic difference in orientation of the ubiquitin between the two structures. Perhaps an alternative view or simply representing distances and angles between the two ubiquitins may help here. Secondly, as the authors acknowledge, this is compared to a crystal structure where lattice constraints may position the ubiquitin in a different conformation. Given the presence of an inhibitor and/or crystal packing may alter the conformation of the ubiquitin the authors may wish to tone down the paragraph on page 5.

We have prepared a new figure (Extended Data Fig. 3a,b) that we believe improves the comparison. Three different orientations of the structural alignment are now provided. The leftmost comparison was chosen to highlight differences in the orientation angle of the ubiquitins (63° about ubiquitin's

central β -sheet), whereas the middle panel shows the magnitude of translation for two distinct secondary structure elements.

To further address these comments, we have edited the relevant paragraph in the Results section comparing the donor ubiquitin positions between our structure and the crystal structure with an inhibitor. Our goal had been to illustrate that the comparison likely identifies the mechanism of action of the inhibitor (by holding the donor ubiquitin in a closed conformation that is distinct from the activated one), which to our knowledge hasn't been described in the literature to date. We have tried to better clarify this in the revised manuscript and thank the reviewer for pointing this out.

In figure 2 the authors make mutations in the synergy loop to reveal the contribution of RBX1 to enhance the activity of UBE2R1 chain elongation activity. On the whole, mutations within the synergy loop impair K_m for unanchored acceptor ubiquitination. However, the greater mutations were from an interface on ubiquitin (R74E). Could this be due to a general impairment of UBE2R1 loading with ubiquitin? In addition, regarding the UBE2R1-interface would an E108R mutation show a comparable defect compared to RBX1 R91E?

The reviewer is astute in pointing out that the R74E mutation in donor ubiquitin may affect E2 activity, as a mild defect in ubiquitylation upon Arg74 chemical modification had been reported as early as 1987 (Duerksen-Hughes et al, *Biochemistry*). Subsequent structural studies have rationalized this result since Arg74 was shown to contact the E1 during ubiquitin activation (Olsen et al, *Molecular Cell*, 2013). Fortunately, this modest effect can be overcome by increasing the E1 concentration to achieve WT-like loading of the donor ubiquitin onto UBE2R1 and UBE2R2 (please note that, except in the single case of BRD4 kinetics, all donor ubiquitins in this study harbor a K48R mutation to suppress its potential for acting as an acceptor ubiquitin). We first show UBE2R2~ubiquitin levels upon E1 titration using a Coomassie-stained gel which enables visualization of both apo E2 and UBE2R2~ubiquitin conjugates. Loading experiments were also performed with radiolabeled ubiquitin and E2~ubiquitin levels were quantified by autoradiography (shown in the graph below the Coomassie-stained gel). These results are now reported in the revised manuscript (Extended Data Fig. 4e).

Regarding the reviewer's second point, it was observed that the E108A UBE2R2 mutant was substantially more defective in k_{obs} (3.5-fold) in comparison with a neddylated CRL^{FEM1C} complex harboring a mutant R91E RBX1 subunit (1.8-fold). This can be rationalized by the high-resolution structure showing that, in addition to its interaction with R91 on RBX1, Glu108 is in proximity with synergy loop residue Arg113. Thus, its mutation may be expected to affect more than UBE2R2 interaction with the RBX1 RING domain. As may be expected of a residue with multiple structural roles, introducing a charge-swapped mutation (E108R) resulted in even greater defects in k_{obs} (8.8-fold; please see Table 1 in the revised manuscript). This point has been addressed in the revised manuscript by updating Fig. 2b showing Glu108's proximity with Arg113.

The end of line 61 should have a reference to the 69 F-box proteins.

We thank the reviewer for pointing this out. We thought it best to delete the number to make the statement more general to other organisms.

Decision Letter, first revision:

Message: Our ref: NSMB-A48077A

1st Dec 2023

Dear Professor Kleiger,

Thank you for submitting your revised manuscript "Mechanism of millisecond Lys48-linked poly-ubiquitin chain formation by cullin-RING ligases" (NSMB-A48077A). It has now been seen by the original referees and their comments are below. The reviewers find that the paper has improved in revision, and therefore we are happy to accept it in principle in Nature Structural & Molecular Biology, pending minor revisions to satisfy the referees' final requests and to comply with our editorial and formatting guidelines.

To facilitate our work at this stage, it is important that we have a copy of the main text as a word file. If you could please send along a word version of this file as soon as possible, we would greatly appreciate it; please make sure to copy the NSMB account (cc'ed above).

Sincerely,

Dimitris Typas
Associate Editor
Nature Structural & Molecular Biology
ORCID: 0000-0002-8737-1319

Reviewer #1 (Remarks to the Author):

The authors have addressed all my concerns and provided additional structural data to strengthen their findings. I support the publication of this exciting work.

Reviewer #2 (Remarks to the Author):

I am satisfied that the authors have addressed all the points that were raised and i recommend the manuscript be accepted for publication. The revised manuscript is much improved and I am pleased to see that the importance of the findings of this study are now better presented in the text.

The only minor point is in the figure legend for Extended Data Figure 4. "e" is missing

from the legend and "f" and "g" are incorrectly assigned.

Reviewer #3 (Remarks to the Author):

The authors have done an excellent job of addressing comments from this reviewer and the two other reviewers.

Author Rebuttal, first revision:

Reviewer #2:

Remarks to the Author:

I am satisfied that the authors have addressed all the points that were raised and I recommend the manuscript be accepted for publication. The revised manuscript is much improved and I am pleased to see that the importance of the findings of this study are now better presented in the text.

The only minor point is in the figure legend for Extended Data Figure 4. "e" is missing from the legend and "f" and "g" are incorrectly assigned.

We thank the reviewer for their comment. The Extended Data Figure 4 legend has been corrected in the revised manuscript.

Final Decision Letter:

Message 21st Dec 2023

:

Dear Professor Kleiger,

We are now happy to accept your revised paper "Mechanism of millisecond Lys48-linked poly-ubiquitin chain formation by cullin-RING ligases" for publication as an Article in Nature Structural & Molecular Biology.

As soon as your article is published, you can generate your shareable link by entering the DOI of your article here: `http://authors.springernature.com/share`. Corresponding authors will also receive an automated email with the shareable link

Your paper will be published online soon after we receive proof corrections and will appear in print in the next available issue. You can find out your date of online publication by contacting the production team shortly after sending your proof corrections.

If you have not already done so, we strongly recommend that you upload the step-by-step protocols used in this manuscript to the Protocol Exchange. Protocol Exchange is an open online resource that allows researchers to share their detailed experimental know-how. All uploaded protocols are made freely available, assigned DOIs for ease of citation and fully searchable through nature.com. Protocols can be linked to any publications in which they are used and will be linked to from your article. You can also establish a dedicated page to

collect all your lab Protocols. By uploading your Protocols to Protocol Exchange, you are enabling researchers to more readily reproduce or adapt the methodology you use, as well as increasing the visibility of your protocols and papers. Upload your Protocols at www.nature.com/protocolexchange/. Further information can be found at www.nature.com/protocolexchange/about.

Please note that *Nature Structural & Molecular Biology* is a Transformative Journal (TJ). Authors may publish their research with us through the traditional subscription access route or make their paper immediately open access through payment of an article-processing charge (APC). Authors will not be required to make a final decision about access to their article until it has been accepted. [Find out more about Transformative Journals](https://www.springernature.com/gp/open-research/transformative-journals)

Authors may need to take specific actions to achieve [compliance with funder and institutional open access mandates](https://www.springernature.com/gp/open-research/funding/policy-compliance-faqs). If your research is supported by a funder that requires immediate open access (e.g. according to [Plan S principles](https://www.springernature.com/gp/open-research/plan-s-compliance)) then you should select the gold OA route, and we will direct you to the compliant route where possible. For authors selecting the subscription publication route, the journal's standard licensing terms will need to be accepted, including [self-archiving policies](https://www.springernature.com/gp/open-research/policies/journal-policies). Those licensing terms will supersede any other terms that the author or any third party may assert apply to any version of the manuscript.

Sincerely,

Dimitris Typas
Associate Editor
Nature Structural & Molecular Biology
ORCID: 0000-0002-8737-1319